# Unusual *Hemiaulus* Bloom Influences Ocean Productivity in Northeast U.S. Shelf Waters

S. Alejandra Castillo Cieza[1], Rachel H.R. Stanley[1*], Pierre Marrec[2], Diana N. Fontaine[2], E. Taylor Crockford[3], Dennis J. McGillicuddy Jr. [3], Arshia Mehta[1], Susanne Menden-Deuer[2], Emily E. Peacock[3], Tatiana A. Rynearson[2], Zoe O. Sandwith[3.4], Weifeng (Gordon) Zhang[3], and Heidi M. Sosik[3]

[1]Chemistry Department, Wellesley College, Wellesley, 02481, USA
[2]Graduate School of Oceanography, University of Rhode Island, Narragansett, 02882, USA
[3]Woods Hole Oceanographic Institution, Woods Hole, MA, 02543, USA
[4]Now at the Hakai Institute, Pruth Harbour, Calvert Island, BC, Canada

*Correspondence to*: Rachel H. R. Stanley (rachel.stanley@wellesley.edu)

**Abstract.** Because of its temperate location, high dynamic range of environmental conditions, and extensive human activity, the long-term ecological research site in the coastal Northeastern U.S. Shelf (NES) of the Northwestern Atlantic Ocean offers an ideal opportunity to understand how productivity shifts in response to changes in planktonic community composition. Ocean production and trophic transfer rates, including Net Community Production (NCP), Net Primary Production (NPP), Gross Oxygen Production (GOP), and microzooplankton grazing rates are key metrics for understanding marine ecosystem dynamics and associated impacts on biogeochemical cycles. Although small phytoplankton usually dominate phytoplankton community composition and Chl-a concentration in the NES waters during the summer, in August 2019, a bloom of the large diatom genus *Hemiaulus*, with $N_2$ fixing symbionts, was observed in the mid-shelf region. NCP was 2.5 to 9 times higher when *Hemiaulus* dominated phytoplankton carbon compared to NCP throughout the same geographic area during the summers of 2020–2022. The *Hemiaulus* bloom in summer 2019 also coincided with higher trophic transfer efficiency from phytoplankton to microzooplankton, higher GOP and NPP, than in the summers 2020-2022. This study suggests that the dominance of an atypical phytoplankton community that alters the typical size distribution of primary producers can significantly influence productivity and trophic transfer, highlighting the dynamic nature of the coastal ocean. Notably, summer 2018 NCP levels were also high although the size distribution of Chl-a was typical and an atypical phytoplankton community was not observed. A better understanding of the dynamics of the NES in terms of biological productivity is of primary importance, especially in the context of changing environmental conditions due to climate processes.

## 1 Introduction

Oceans regulate atmospheric carbon dioxide ($CO_2$) concentrations and support life on Earth via several mechanisms (Friedlingstein et al., 2022). One of these mechanisms is the biological pump, which involves biological, physical, and chemical processes that aid in transporting and sequestering organic carbon from $CO_2$ (Boyd et al., 2019). As the main primary producers in the ocean, phytoplankton play a major role in the biological pump (Field et al., 1998). Diatoms, a type of photosynthetic algae, are believed to account for nearly half of net marine primary productivity globally and are important contributors to the biological pump (Jin et al., 2006). Diatoms characteristically thrive in nutrient-rich surface layers and turbulent conditions, and are thus typically found at high latitudes and in coastal upwelling regions (Armbrust, 2009). However, new technology (e.g., molecular biology and imaging) has revealed that diatoms may be more prevalent in low nutrient, oligotrophic systems than traditionally considered (Malviya et al., 2016), likely due to unique metabolic capabilities involving nutrient acquisition strategies that enable their survival in low nutrient regimes (Margalef, 1978).

One specific metabolic capability within diatoms is the ability to form a symbiosis with nitrogen-fixing cyanobacteria. This symbiosis, known as a diatom-diazotroph association, has been observed around the globe, mostly in oligotrophic regions (Foster and Zehr, 2019), but also in temperate continental shelf waters (Wang et al., 2021). Furthermore, some diatom-diazotroph association have the capability to grow very quickly, forming localized

blooms (Villareal et al., 2011). Diatom-diazotroph blooms, specifically involving the diatom genus *Hemiaulus* and
the symbiont *Richelia*, have been found in warm, stratified waters in various regions around the globe and have been
associated with high carbon export observed via a combination of modern oceanographic measurements and paleo-
flux case studies. Examples include blooms in the eastern Equatorial Atlantic (Foster and Zehr, 2006), tropical North
Atlantic (Carpenter et al., 1999; Subramaniam et al., 2008), North Pacific Subtropical Gyre (Dore et al., 2008;
Villareal et al., 2011), and South China Sea (Grosse et al., 2010). Furthermore, at the ALOHA site in the Pacific
Ocean north of Hawaii, blooms of the *Hemiaulus-Richelia* association can last as long as 30 days and contribute
significantly (20%) to annual carbon flux in this region (Kemp and Villareal, 2018; Karl et al., 2012). Diatoms with
nitrogen-fixing symbionts are thus important contributors to primary productivity and carbon export, especially at
times when surface waters are depleted of dissolved inorganic nitrogen (Tang et al., 2020).
An intense bloom of *Hemiaulus* and its symbiont *Richelia* was observed in summer 2019 in temperate
Northeast U.S. Shelf (NES) surface waters. The NES region in the Northwestern Atlantic Ocean is particularly
productive, favoring enhanced inorganic carbon sequestration by the biological pump, and supports an ecologically
and economically important ecosystem (Townsend et al., 2006). Like other marine regions, the NES food web is
fueled by phytoplankton, the main primary producers, which play a fundamental role in the ecosystem (e.g. Mouw
and Yoder, 2005; O'reilly and Zetlin, 1998; Yoder et al., 2002). Productivity is heavily influenced by abiotic factors
in the NES region. For instance, strong seasonal variations in water temperature, stratification and cross-shelf
advection on the NES affect nutrient supply and lead to seasonal shifts in phytoplankton productivity and species
composition (Li et al., 2015; Oliver et al., 2022; Zhang et al., 2023). Furthermore, the water temperature of the NES
is rising faster than the global average (Chen et al., 2020; Karmalkar and Horton, 2021; Shearman and Lentz, 2010),
leading to unknown consequences for phytoplankton community composition and productivity within this important
and dynamic coastal region.
To further understand phytoplankton population dynamics and their influence on the ocean's biological
pump, the NES Long-Term Ecological Research (NES-LTER, https://nes-lter.whoi.edu/) project investigates
primary productivity, food web structure and ecosystem dynamics with a focus on southern New England coastal
waters. As part of the NES-LTER project, phytoplankton and zooplankton community composition, phytoplankton
growth rates, microzooplankton grazing rates, and productivity rates are determined on week-long research cruises
which have occurred quarter-annually since 2018. To quantify productivity, several different rates are estimated
from data collected on these cruises, including Gross Oxygen Production (GOP), Net Primary Production (NPP),
Net Community Production (NCP) and export efficiency ratios (NCP/GOP). GOP is similar to Gross Primary
Production; it represents total photosynthesis in oxygen units and also includes photoprocesses that produce oxygen
(Juranek and Quay, 2013). NPP is photosynthetic production minus autotrophic respiration and thus represents the
net production activity of the phytoplankton community. NCP is the balance of photosynthesis and community
respiration (autotrophic plus heterotrophic) and is equal, on long enough spatial and temporal scales, to the amount
of carbon exported out of the surface of the ocean (Emerson, 2014). The NCP/GOP ratio, analogous to the f-ratio
(Dugdale and Goering, 1967), is indicative of export efficiency, with a high ratio implying that the community is
exporting most of the carbon (organic matter) produced and thus recycling only a little (Juranek and Quay, 2013).
The composition and size structure of the phytoplankton community in the NES-LTER study are
investigated concurrently from automated imaging and size-fractionated chlorophyll-a (Chl-a). In winter, the NES
waters tend to be nutrient-rich due to enhanced vertical mixing and input of river and estuary waters that promote
high levels of surface Chl-a, with a dominance of large phytoplankton cells that grow slowly (Marrec et al., 2021).
Conversely, during a typical summer, nutrients become depleted in the surface mixed layer, leading to low Chl-a
concentrations dominated by fast-growing small phytoplankton cells (Marrec et al., 2021; O'reilly and Zetlin,
1998).
To complement production estimates and phytoplankton community structure observations, the flow of
carbon from primary producers to higher trophic levels on the NES has also been investigated. Microzooplankton,
protists smaller than 200 μm, are a crucial link between primary producers and higher trophic levels because they
often consume 60–70% of daily primary production (Landry and Calbet, 2004; Schmoker et al., 2013). In the NES,
while phytoplankton grow faster during the summer than in winter, microzooplankton grazing rates tend to stay
relatively constant across seasons (Marrec et al., 2021). Thus, during winter, phytoplankton growth rates and
microzooplankton grazing rates are typically well coupled and show a close 1:1 ratio, with microzooplankton
consuming most of the primary production (Marrec et al., 2021). During the summer, phytoplankton growth and
microzooplankton grazing rates are typically decoupled, leading to less than 50% of the primary production
consumed by microzooplankton. The degree of coupling between microzooplankton grazing and phytoplankton
growth rates is associated with phytoplankton size structure (Marrec et al., 2021) and likely species composition,
and is an important indicator of the trophic transfer efficiency from phytoplankton to microzooplankton at the base
of the planktonic food web.
Here, we examined the association between productivity, phytoplankton composition and
microzooplankton grazing, key components of trophic transfer efficiency and ecosystem function. During a NES-
LTER cruise in summer 2019, we observed an anomalous relationship between growth and grazing rates, as well as
dramatically different productivity rates and community composition compared to other summer cruises in the NES
region. We thus investigated how a diatom bloom of *Hemiaulus,* with diazotrophic symbionts, affected metrics of
productivity and grazing on the NES during the summer of 2019. Our results provide insights into the effects of
community composition on productivity rates.
**2 Methods**
Measurements of environmental conditions, chemical and biological stocks, and productivity and grazing rates were
conducted on multiple cruises within the framework of the NES-LTER program (Table 1). Measurements from three
other cruises from different projects on the NES were also included in this analysis for comparison (project names in
Table 1) and *Hemiaulus* abundances were further compared to an additional 26 cruises in the NES (Table S1). From
the time series, we were able to better understand an event that was observed on the 2019 NES-LTER summer
cruise (EN644) which occurred August 20 to 25 (Table 1). Some data during that event, such as surface seawater
temperature (SST), salinity (SSS), NCP rates, and phytoplankton composition were collected continuously from the
underway system (every 0.1 to 6 km depending on the measurement type and ship speed), while other parameters
(e.g., NPP, grazing rates, Chl-a, nutrients) were measured discretely at the NES-LTER stations (Fig. 1, Table S2).
Main stations were located with ~ 19 km spacing on a north-to-south transect primarily along 70.883º W. Fig. 1
shows the cruise track for the August 2019 NES-LTER cruise, but all the other NES-LTER cruises had a near
identical cruise track. In particular, the mid-shelf region, which is where the *Hemiaulus* bloom primarily occurred,
corresponds to 50 – 100 m water depth and was bounded by latitudes 40.980 ºN to 40.327 ºN. The mid-shelf region
contains four stations. Exact locations and dates of when the mid-shelf stations were occupied is provided in Table
S2.
At each station, water was collected via Niskin bottles mounted on a CTD-rosette (conductivity-
temperature-depth, Seabird SBE32 Carousel Water Sampler). The CTD-rosette system consisted of a 24-bottle
rosette frame with 10-L Niskin bottles. Depth, temperature, and salinity were collected with a SBE911 CTD
(Seabird Electronics) equipped with additional sensors for chlorophyll fluorescence (WET Labs ECO-AFL/FL),
photosynthetically active radiation (PAR, Biospherical Instruments® QSP2000), and beam attenuation (WET Labs
C-Star 25-cm transmissometer). The Niskin bottles were closed at various depths ranging from surface to near
bottom, based on the depths of the mixed layer, euphotic zone, and Chl-a maximum. Mixed layer depths were
calculated from the temperature and salinity data from the CTD with the threshold method where the mixed layer
was taken to be the depth where the density difference between the surface and bottom of the mixed layer was
greater than $\Delta\sigma_\theta = 0.125$ kg m$^{-3}$ (De Boyer Montegut et al., 2004). Mixed layer depths were confirmed to be similar
when a gradient criterion with a difference of 0.0125 kg m$^{-3}$ was used instead (Kara et al., 2000). Euphotic Zone was
taken to be the depth at which light was 1% of the surface value. Chl-a max was chosen based on the depth with
maximum fluorescence observed in the CTD cast. Water from the Niskins was used to quantify a number of
parameters as described in Sections 2.2 through 2.5.
The underway system consisted of continuous surface seawater pumped throughout the ship by an impeller pump
and a diaphragm pump located near the ship's bow. Using water from the impeller pump, continuous measurements
of surface temperature and salinity were obtained from a Seabird SBE38 (temperature) sensor installed at the water
intake and by a Seabird SBE45 sensor (temperature and salinity) located further away in the underway system.
Because the diaphragm pump is less likely to damage plankton (Cetinic et al., 2016), its underway flow was used for
measurements to quantify NCP (Section 2.1), GOP (Section 2.2), and phytoplankton community composition
(Section 2.8). The ship steamed both south and north along the longitude 70.883°W and thus over the 6-day cruise,
the underway data sampled the same locations at multiple points in time. Stations were only occupied at one time
per cruise.


.

**Table 1.** Dates of the summer cruises, as well as project and ship names and cruise numbers, that are presented in
this paper. Project name abbreviations are as follows: OTZ– Ocean Twilight Zone, SPIROPA–Shelfbreak
Productivity Interdisciplinary Research Operation at the Pioneer Array (Oliver et al., 2021), and EcoMon–
Ecosystem Monitoring program run by the National Oceanic and Atmospheric Administration. Cruise tracks for the
NES-LTER transects are shown in Fig. 1. The SPIROPA and OTZ cruises followed the same longitude 70.883°'W
when in the mid-shelf region and thus data used from those cruises is collocated with the NES-LTER data.

| Cruise | Start date/End date | Project name | Ship |
|--------|--------------------|--------------|------|
| EN617 | 20 July 2018 – 25 July 2018 | NES-LTER | *R/V Endeavor* |
| TN368 | 05 July 2019 – 18 July 2019 | SPIROPA | *R/V Thomas G. Thompson* |
| HB1907 | 25 July 2019 – 08 Aug 2019 | OTZ | *NOAA Ship Henry B Bigelow* |
| GU1902 | 16 Aug 2019 – 29 Aug 2019 | EcoMon | *NOAA Ship Gordon Gunter* |
| EN644 | 20 Aug 2019 – 25 Aug 2019 | NES-LTER | *R/V Endeavor* |
| EN655 | 25 July 2020 – 28 July 2020 | NES-LTER | *R/V Endeavor* |
| EN668 | 16 July 2021 – 21 July 2021 | NES-LTER | *R/V Endeavor* |
| EN687 | 29 July 2022 – 03 Aug 2022 | NES-LTER | *R/V Endeavor* |


## 2.1 Net Community Production

Net community production rates were calculated from $O_2$/Ar ratios measured by an at-sea Equilibrator Inlet
Mass Spectrometer (EIMS) (Cassar et al., 2009) analyzing water from the ship's underway system and from discrete
samples collected from both CTD Niskin bottles and from the underway system. The EIMS was used to collect
continuous data on $O_2$/Ar ratios via the diaphragm pump of the underway system that, on the *R/V Endeavor*, pumps
seawater from a depth of 5 m. The underway system seawater flows through a debubbler into a bucket at a constant
rate that allows for continuous overflow for consistent head pressure. Water is then pumped from the bucket at ~1.1
L min$^{-1}$ by a gear pump through two filters: a bag with a 25-µm pore size, and a 2-layered sock with a 5-µm inner
and 100-µm outer pore size. The gear pump then pushes the water through an equilibrator membrane contactor
cartridge (Liqui-Cel Extra-Flow 2.5x8 model G540). The equilibrated headspace gas from the cartridge is then dried
by flowing through the dessicants Nafion and Drierite and then passed via a fused silica capillary into a Hiden
Residual Gas Analyzer (RGA) (HAL 7) quadrupole mass spectrometer. Details of the equilibration method can be
found in Manning et al. (2016), but in this instance were modified to not use SAES getters as they would have
removed the $O_2$. The EIMS was operated throughout the whole cruise (starting one hour after the ship left port and
ending a few hours before return to port). To calibrate the mass spectrometer, the capillary was switched to an air
inlet for twenty minutes approximately every six hours as the ratio of $O_2$/Ar in air is stable and well-known.
Additionally, bottle samples were collected from the underway system at least once per day and were subsequently
measured on an isotope ratio mass spectrometer at Woods Hole Oceanographic Institution (see Section 2.2). These
bottle samples were used to provide additional calibration as necessary–such additional corrections changed the
$O_2$/Ar ratios by at most 0.67%.
The $O_2$/Ar ratios were then used to calculate NCP (Hendricks et al., 2004; Juranek and Quay, 2005; Stanley
et al., 2010). With data from the EIMS and the bottle samples, the biological oxygen saturation $\Delta(O_2/Ar)$ was
calculated via the equation below:
$$\Delta\left(\frac{O_2}{Ar}\right) = \frac{\left(\frac{O_2}{Ar}\right)_{smpl}}{\left(\frac{O_2}{Ar}\right)_{eq}} - 1 \tag{1}$$


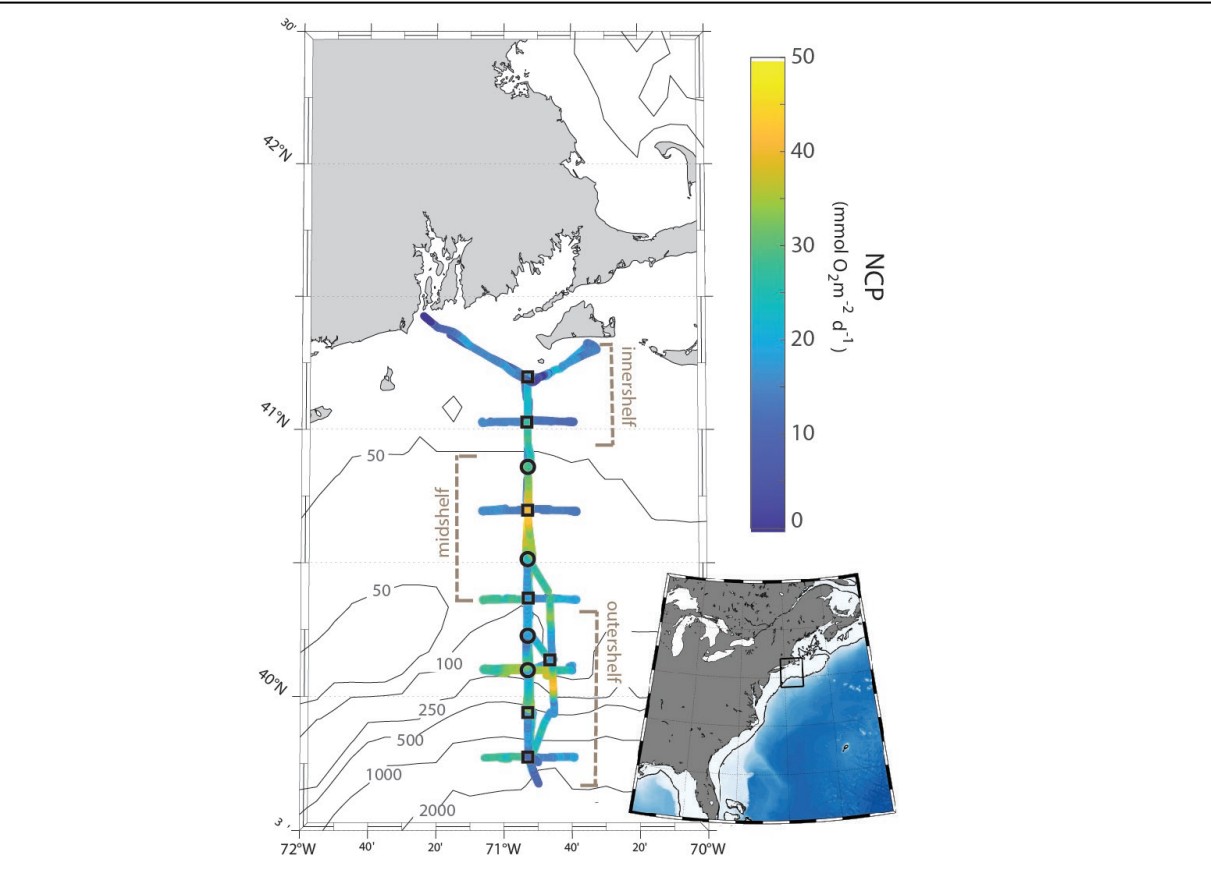

**Fig 1.** Map of the NES-LTER August 2019 cruise track colored according to rates of NCP as measured continuously by an at-sea mass spectrometer for the second half of the cruise. Station locations are marked with circles or squares and are ~19 km apart. Rates of NCP, GOP, and phytoplankton community composition were quantified at all stations. Grazing rates and NPP in 2019 to 2022 were calculated only at stations marked by squares. Other NES-LTER cruises have a similar track, although issues such as weather sometimes change the track slightly and in 2018. The inset shows the location of the cruise (rectangular box) in the context of the east coast of the United States.

where $(O_2/Ar)_{smpl}$ represents the ratio of $O_2$ to Ar ion currents detected by the EIMS after being calibrated with
bottle data, and $(O_2/Ar)_{eq}$ represents the ratio of equilibrium concentrations of the gases determined from the gases'
solubility (Garcia and Gordon, 1992; Hamme and Emerson, 2004) at the seawater temperature and salinity.
The NCP integrated over the mixed layer, in units of mmol $O_2$ m$^{-2}$ d$^{-1}$, is calculated as

$$NCP = \Delta\left(\frac{O_2}{Ar}\right)[O_2]_{eq}k\rho \qquad (2)$$

where $[O_2]_{eq}$ represents the equilibrium concentration of $O_2$ at the relevant temperature and salinity (mmol kg$^{-1}$), $k$ is
the weighted gas transfer velocity (m d$^{-1}$), and $\rho$ is the density of seawater (kg m$^{-3}$) (Millero and Poisson, 1981). The
weighted gas transfer velocity is a time-weighted average from over the past 30 days calculated as described in
Reuer et al. (2007), with the gas exchange parameterization of Stanley et al., (2009) and wind speeds from NCEP
Reanalysis (Kalnay et al., 1996; Kistler et al., 2001). Many physical considerations altering $O_2$ saturations, such as
changes in temperature and bubble injection, do not need to be considered due to the inclusion of Ar which has
similar solubility and diffusivity as $O_2$; however, a few assumptions were made for these calculations. Firstly, this
equation assumes steady state within the mixed layer, i.e. no change in $O_2/Ar$ in the ocean with time. While $O_2/Ar$
was likely changing in actuality, assuming steady state simply means that the rates calculated reflect an
exponentially weighted average of NCP over the past few residence times of oxygen (residence time equals a few
days in these conditions) (Teeter et al., 2018). Thus, the assumption of steady state does not majorly impact our
conclusions. We were not able to calculate the time rate of change term in $O_2/Ar$ (Manning et al., 2017b) because
the cruise was not Lagrangian, and even though the ship returned to the same geographic location, the water at that
location changed due to ocean currents. To check the assumption that there is negligible respiration within the ship's
lines (Juranek et al., 2010), bottle samples were collected in duplicate from Niskins at the same time as samples
were collected from the underway system several times during every cruise; gas concentrations in the bottle samples
from the underway and Niskin were identical within measurement errors, confirming there was no detectable
respiration in the ship's line.

## 2.2 Gross Oxygen Production

Discrete samples of triple oxygen isotopes (TOI) were collected from the surface Niskin bottles on the CTD-rosette
system at all stations as well as from the underway system between stations. Samples from the CTD-rosette system
were also collected from bottles fired at $\sim 5$ m below the mixed layer and often one greater depth to provide
information for assessing whether vertical corrections to $O_2/Ar$ ratios were significant. Samples were collected in
custom-made $\sim 500$-mL sample bottles which were pre-poisoned with 100 µl of saturated mercuric chloride solution
and filled with around 300 mL of seawater from the underway system or from the Niskin at each station (Stanley et
al., 2015). Samples were brought to Woods Hole Oceanographic Institution where they were analyzed for TOI with
a custom-made processing line and a Thermofisher MAT 253 isotope ratio mass spectrometer as detailed in Stanley
et al (2015). The same samples were also analyzed for $O_2/Ar$ which yielded rates of NCP from discrete data as well
as an independent method for calibrating the EIMS (see above). Corrections for the effect of argon on the triple
oxygen isotope ratio and the effect of varying sizes of the sample vs. reference standard were made for every
sample. Reproducibility from duplicate samples collected on these cruises ranged from 4 to 8 per meg for $^{17}\Delta$, 0.008
to 0.03 per mil for $\delta^{17}O$, and 0.008 to 0.05 per mil for $\delta^{18}O$ depending on the cruise.
From these samples, GOP is calculated in units of mmol $O_2$ $m^{-2}$ $d^{-1}$ following Prokopenko et al., (2011)
according to:
$$GOP = kO_{eq} \frac{\frac{x_{dis}^{17}-x_{eq}^{17}}{x_{dis}^{17}} - \lambda \frac{x_{dis}^{18}-x_{eq}^{18}}{x_{dis}^{18}}}{\frac{x_{P}^{17}-x_{eq}^{17}}{x_{dis}^{17}} - \lambda \frac{x_{P}^{18}-x_{eq}^{18}}{x_{dis}^{18}}} \qquad (3)$$

where $k$ again represents the time-weighted gas transfer velocity (m $d^{-1}$), $O_{eq}$ represents the equilibrium
concentration of oxygen, $\lambda$ represents the respiration slope factor = 0.5179, $X_{dis}*$ represents the ratio of isotopes
($*O/^{16}O$) dissolved in the sample, $X_{eq}*$ represents the ratio of isotopes ($*O/^{16}O$) dissolved in seawater equilibrated
with the atmosphere, and $X_P*$ stands for the ratio of isotopes ($*O/^{16}O$) in oxygen that was produced via
photosynthesis. The photosynthetic end member used was the average of the phytoplankton value determined by
Barkan and Luz (2011); Vienna Standard Mean Ocean Water (VSMOW) was used for the isotopic composition of
oxygen in $H_2O$. The actual isotopic composition of $H_2O$ was measured in a subset of samples to see if corrections
needed to be made (Manning et al., 2017a). It was found to be very similar to VSMOW, leading to an error of less
than 10% in GOP due to isotopic water variations.
Confirmation that the water from the underway system was representative of the oceanic TOI signature of
dissolved oxygen was obtained by comparing samples collected from the underway system to those collected
concurrently from the surface Niskin bottle. All cruises, other than 2019, showed that there was statistically no
difference in TOI between water from the underway system and the CTD and thus that the water from the underway
system was representative of the mixed layer at that location and time. During the summer of 2019, the water from
the underway system had TOI values 4.1 per meg lower than that from the CTD – this is within measurement errors
but since it might have led to systematic biases, we corrected for this offset before calculating GOP from the data.
The GOP rates, along with the NCP rates, represent productivity integrated throughout the mixed layer.

## 2.3 Net Primary Productivity

Water samples for NPP were collected at 4-7 stations (cruise dependent) from 3-4 depths (station
dependent) from the Niskins on the CTD-rosette system during the summers of 2019 to 2022. During collection,
water was pre-filtered through 200-µm mesh (to remove mesozooplankton) into acid-washed 2-L polycarbonate
bottles. Water collection and associated incubation occurred in triplicate for surface samples at each station. Bottles
were spiked with a solution of 99% $NaH^{13}CO_3$ (Cambridge Isotope Lab, Tewksbury, MA) for a final 10%
enrichment of the dissolved inorganic carbon (DIC) pool and placed in various mesh bags to simulate *in situ* light

levels. Bottles were incubated for 24 h in clear deck-board incubators with flowthrough seawater and Onset HOBO data loggers monitored tank water temperature. At each station, the natural $^{13}C$ in the water was determined from an un-spiked sample and dark carbon assimilation was determined from a spiked dark bottle sample. Dark carbon assimilation was negligible (<1%) so no correction for dark carbon assimilation was applied to this dataset.

The corresponding light levels at collection depths were determined using either PAR or beam attenuation from the CTD cast for each station. When PAR data were not available (e.g., night-time casts), a relationship was established (eq. 4) with previous daytime cast information between beam attenuation ($c$, measured by transmissometer, $m^{-1}$) and the light extinction coefficient ($K_d$, $m^{-1}$) for each cruise. During night-time casts, $K_d$ was estimated from the average $c$ in the upper 10 m during the cast with the slope (m) and intercept (b) from the daytime plot, according to equation 4:

$$K_d = (m * At) + b \qquad (4)$$

The appropriate shading in incubations (%PAR) for each depth of sample collection (z) was estimated as:

$$\%PAR = 100e^{-K_d \times z} \qquad (5)$$

At the end of each incubation, bottles were filtered under low vacuum (5-10 in. Hg) over pre-combusted Whatman GF/F filters (450°C; 6h). Filters were stored at -20°C until further analysis on shore. NPP rates were quantified by measuring the incorporation of isotopically heavy carbon into phytoplankton biomass. Prior to measuring $^{13}C$ in the samples, filters were acid fumigated with concentrated HCl in a desiccator overnight to remove inorganic carbon. They were dried in an oven at 60°C for 24 h, individually wrapped in tin capsules and analyzed on a Carlo Erba NC2500 elemental analyzer interfaced with a Thermo Delta V+ isotope ratio mass spectrometer. The $\delta$ $^{13}C$ values were reported relative to the international standard Vienna PeeDee Belemnite (Coplen, 1995) and converted to atom percent values.

NPP rates were calculated from atom percent values with the equation from Hama et al. (1983)

$$NPP = \frac{POC \cdot (a_{is} - a_{ns})}{t \cdot (a_{ic} - a_{ns})} \qquad (6)$$

where NPP is the net primary production rate ($\mu g \cdot L^{-1} \cdot day^{-1}$), POC is the particulate organic carbon; ($\mu g \cdot L^{-1}$), $t$ is the incubation time (h), $a_{is}$ is the atom % of $^{13}C$ in the incubated sample, $a_{ns}$ is the atom % of $^{13}C$ in the natural sample (un-spiked sample described above) and $a_{ic}$ is the atom % of $^{13}C$ in the total DIC pool. POC measurements were blank corrected with the mean value of triplicate combusted filter blanks. The DIC concentration was determined from salinity (S) according to the following equation from Parsons et al. (1984):

$$DIC = \big((S * 0.067) - 0.05\big) * 0.96 \qquad (7)$$

NPP rates were integrated to the depth of the mixed layer (Table S3) to align with NCP and GOP integrated rate calculations. In summer, NPP rates below 16 m (deepest mixed layer depth) were not used in this study.

No discrete measurements of net primary productivity (NPP) were conducted during the summer of 2018. However, we were able to estimate 2018 NPP rates as follows: For each summer, we computed phytoplankton biomass production (PP, mg C $m^{-3}$ $d^{-1}$) based on surface discrete Chl-a concentration and growth/grazing rates, following the methodology outlined by Landry et al. (2003). Chl-a concentrations were transformed into biomass using a constant C:Chl-a ratio of 50. In the summers from 2019 to 2022, where discrete NPP data were available, we averaged surface PP and surface NPP by region (inner-shelf, mid-shelf, and outer-shelf) and conducted a linear regression between these average PP and NPP rates ($p < 0.05$; $R^2 = 0.68$; $\cdot$ n= 15). The linear regression coefficient obtained from this correlation was used to convert PP derived from growth/grazing rates in the summer of 2018 into NPP (mg C $m^{-3}$ $d^{-1}$). Subsequently, we integrated NPP over the mixed layer to obtain integrated NPP (mg C $m^{-2}$ $d^{-1}$) at each station where surface growth/grazing rates were available. While the C:Chl ratios in coastal systems exhibit high seasonal variability (Jakobsen and Markager, 2016), we used a constant C:Chl ratio when converting Chl-a into phytoplankton biomass. Since our comparison of derived PP was limited to the summer season, it is reasonable to assume that C:Chl ratios remained within a similar range. Additionally, the same C:Chl ratio was used when deriving the linear relationship and when applying it and thus the estimated NPP rates are insensitive to the choice of C:Chl ratio. It is important to note that C:Chl ratios were not utilized in the calculation of NPP rates for any other year.

**2.4 Autotrophic and Heterotrophic Respiration**

Assuming a photosynthetic quotient (O:C ratio) of 1.4, respiration rates were calculated from the productivity values (GOP, NPP, and NCP) and following the relationships below:

$$NPP = GOP - R_A \qquad (8)$$
$$NCP = NPP - R_H \qquad (9)$$

where $R_A$ is autotrophic respiration and $R_H$ is heterotrophic respiration.

## 2.5 Growth Rates and Grazing rates

Rates of phytoplankton growth and protistan grazing were quantified with a 2-point modification of the dilution method (Landry et al., 2008; Chen, 2015; Morison et al., 2020) following methods in Marrec et al. (2021). Briefly, surface samples were collected at 4 to 7 stations throughout the cruise. For each sample, whole seawater (WSW) from the Niskin bottles was transferred into a 10-L polycarbonate carboy through a 200-μm mesh filter to remove mesozooplankton predators. Diluent was prepared by gravity filtration through a 0.2 μm membrane filter capsule (PALL®) from the Niskin to the carboys and mixed with WSW to obtain a 20% WSW dilution. A total of 6 bottles per experiment were prepared: 2 bottles with nutrient amended 20% WSW, 2 bottles with nutrient-amended WSW, and 2 bottles with unamended WSW to assess nutrient limitation. Incubations took place for 24 h in a clear, 1m³ deck-board incubator. Paired bottles were placed into mesh bags that simulated the effective light availability in the surface mixed layer, which corresponded to 65% of sea surface irradiance. Phytoplankton growth and grazing mortality rates were then estimated from changes in Chl-a over the 24 h incubation. For dilution experiments, Chl-a concentrations were obtained from triplicate 150-mL subsamples filtered on GF/F filters, after a 12-h dark extraction period at room temperature in 95% ethanol and measured on a calibrated Turner 10 AU fluorometer. The full extraction method is detailed in Marrec et al. (2021).

## 2.6 Discrete Chlorophyll-a sample collection and processing

Samples for Chl-a analysis were collected into brown amber bottles from Niskins on the CTD Rosette system. A known sample volume (250-500 mL) was filtered at low pressure (5-10 in. Hg) through either a GF/F filter or a 20 μm polycarbonate Sterlitech filter. Filters were transferred to either tissue capsules (GF/F) or cryogenic vials (20 μm) and then flash frozen in liquid nitrogen until extraction. Later, filters were extracted in 5 mL of 90% acetone for 24 hours in a dark refrigerator, then tubes were vortexed and centrifuged (only GF/F filters), and the solution was measured on a calibrated Turner Designs Handheld Aquaflor fluorometer, acidified with 2 drops of 10% hydrochloric acid and measured again. Chl-a concentrations for different size fractions were calculated by difference. Note that in this study we consider large phytoplankton are as > 20 μm.

## 2.7 Satellite and radar data

To look at variability in SST and surface Chl-a, a proxy for phytoplankton biomass, throughout the summers over multiple years, on a wider spatial and temporal scale than the at-sea chlorophyll data permitted, SST and surface Chl-a concentrations from remote sensing sources were retrieved and analyzed. In particular, both snapshots and monthly averages of MODIS (Moderate Resolution Imaging Spectroradiometer) SST and chlorophyll data with a horizontal resolution of 1 km were used to examine the spatial coverage of the *Hemiaulus* bloom in summer 2019 (when it dominated phytoplankton biomass) and compare the surface temperature and chlorophyll in the NES region in summers 2018-2022.

To examine possible origins of the bloom water, backward particle trajectory simulations were carried out with the OceanParcels Python package https://oceanparcels.org/index.html (Lange and Van Sebille, 2017). High frequency (HF) radar-measured sea surface velocity data in the NES region in Jul-Aug 2019 with 6-km spatial resolution and hourly temporal resolution were used as the background flow. Particles were released at mid-shelf sites along the NES-LTER transect on Aug 21, 2019 and advected backward for 30 days until Jul 22, 2019.

## 2.8 Imaging FlowCytobot

Composition of the phytoplankton community was assessed with Imaging FlowCytobots (IFCB; McLane Research Laboratories, Inc.). IFCB uses a combination of video and flow cytometry technology to capture images of plankton and other particles in the size range ~5-150 μm (Olson and Sosik, 2007). During the cruises reported here, IFCB instruments were configured to record images of particles with laser-based chlorophyll fluorescence or light scattering signals above trigger thresholds and samples were pre-screened with 150 μm Nitex. IFCB instruments were operated two ways. First, on all cruises, an IFCB was configured to sample 5 mL automatically from the ship's

underway system every 25 minutes. Second, at stations occupied on the NES-LTER and SPIROPA cruises, IFCB
instruments were used to analyze depth profiles from discrete samples collected with Niskin bottles. Typically, three
5-mL subsamples were measured for each depth. The fraction of each 5-mL sample imaged by IFCB decreases with
increasing trigger rate but is recorded precisely during sample acquisition enabling calculation of concentrations.
IFCB image data were automatically analyzed following approaches developed for the IFCB time series at the
Martha's Vineyard Coastal Observatory (MVCO) (Brownlee et al., 2016). In particular, cell biovolume was
estimated from IFCB images (Moberg and Sosik, 2012) and converted to cell carbon following the relationships
described by Menden-Deuer and Lessard (2000). IFCB images were classified with a convolutional neural network
(CNN) (Catlett et al., 2023) trained to separate 155 categories of plankton and other particles observed at MVCO
and across the NES region. We used the Inception v3 (Szegedy et al., 2016) CNN architecture as implemented in
PyTorch, pre-trained with ImageNet (Russakovsky et al., 2015) and fine-tuned with an NES IFCB training set
(97026 images, 155 classes, 80-20 split for training and validation). In addition, an independent test set of manually
annotated images in 51 IFCB samples from EcoMon cruises was used to evaluate *Hemiaulus* quantification as a
function of classifier score threshold. From this independent analysis, classifier predictions with scores above 0.9
performed very well for *Hemiaulus* (class-specific F1-score = 0.936; CNN-count vs. manual-count: $r^2$ = 0.999, slope
= 0.915; intercept = 0.005).

### 2.9 Nutrients

Dissolved inorganic nutrient concentrations (ammonium, phosphate, silicate, and nitrate + nitrite) were obtained
from CTD bottle samples with duplicates. Seawater was passed through an EMD Millipore sterile Sterivex 0.22 µm
filter with filtrate collected into acid-washed 20-ml scintillation vials (after triplicate rinses), which were then stored
at -20 °C until analysis. Samples were processed at Woods Hole Oceanographic Institution's Nutrient Analytical
Facility with a four-channel segmented flow SEAL AA3 HR Autoanalyzer. Detection levels are as follows: 0.01
µmol L$^{-1}$ for silicate, 0.03 µmol L$^{-1}$ for phosphate, 0.04 µmol L$^{-1}$ for nitrate + nitrite, and 0.03 µmol L$^{-1}$ for
ammonium.

## 3 Results

### 3.1 *Hemiaulus* distribution and Chlorophyll

During the NES-LTER summer 2019 cruise, through automated image classification and analysis and through visual
microscopic confirmation, a bloom of the diatom genus *Hemiaulus* was observed in the surface waters of the mid-
shelf region (Fig. 2a). These images also showed $N_2$ fixing symbionts, namely *Richelia,* inside or next to the

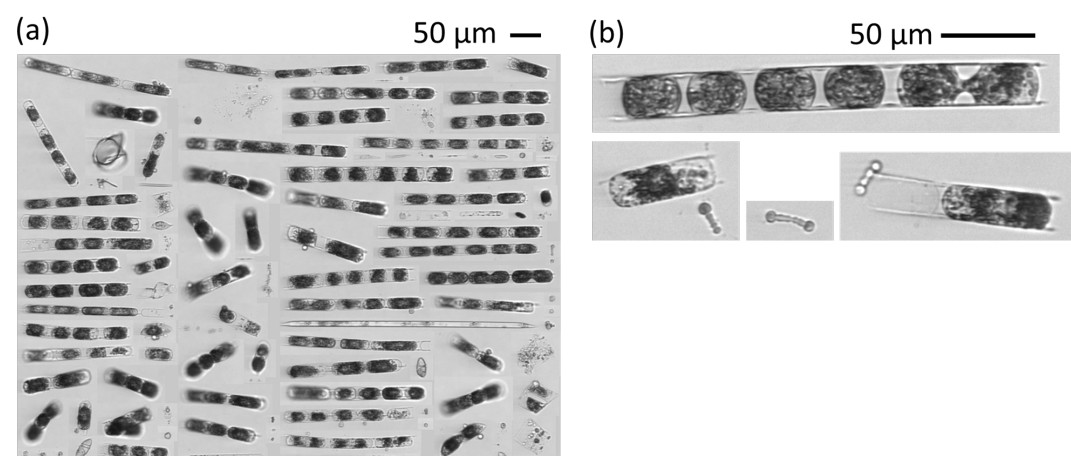

**Fig 2.** IFCB images of the **(a)** phytoplankton community during the summer 2019 NES-LTER cruise and **(b)** closer up individual *Hemiaulus* cells found with the nitrogen-fixing symbiont *Richelia*.

*Hemiaulus* cells (Fig. 2b). Additionally,
*Hemiaulus* carbon was highest in the mid-
shelf waters between latitudes of 40.1° N
and 41.1° N, a span of 111 km (Fig. 3a),
with concentrations ranging from 6.8 to
68.3 µg L$^{-1}$. This bloom was only observed
in the surface waters of the mid-shelf
region, as can be seen by discrete IFCB
measurements from Niskin samples (Fig.
3b). *Hemiaulus* carbon concentrations
observed in other years on NES-LTER
transect cruises never reached values above
0.30 µg L$^{-1}$, so approximately two orders of
magnitude smaller than was observed on
the 2019 cruise. Furthermore, IFCB-based
observations made on a broader scale from
the mid-Atlantic bight to the Gulf of Maine
in the period from 2013 to 2023, show that
only in August 2019 is *Hemiaulus* present
in large quantities (Fig. S1), confirming the
extraordinary nature of the 2019 bloom.
398   The presence of the diatom bloom
was consistent with the size-fractionated
Chl-a data. Surface Chl-a concentrations in
the mid-shelf region in summer are
typically low (< 0.50 µg L$^{-1}$, Fig. 4a) and
progressively decrease with decreasing
latitude. During the NES-LTER summer
2019, however, Chl-a concentrations were
as high as 3.50 µg L$^{-1}$ in the surface waters
of the mid-shelf (mean Chl-a of 1.97 µg L$^{-1}$,
Table 2) with up to 80% of the Chl-a
associated with the > 20 µm fraction (Fig.
4c). This is in contrast to other summers
when most of the Chl-a was associated
with the < 20 µm fraction (Fig. 4b, d-f).
Concentrations of Chl-a in the > 20 µm
size fraction and concentrations of
*Hemiaulus* carbon in the NES-LTER
summer 2019 cruise were larger at co-

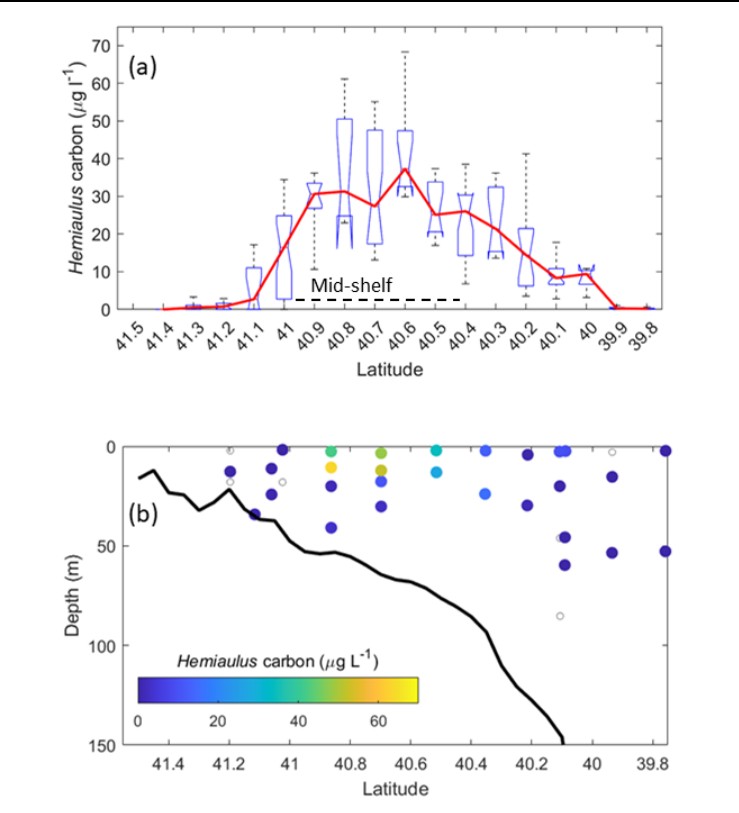

**Fig 3.** Cross-shelf distribution of *Hemiaulus* carbon concentration in August 2019 showing the mixed-layer bloom in the mid-shelf region. Results are derived from IFCB observations in **(a)** surface waters and **(b)** discrete samples from depth profiles with symbols color-coded by *Hemiaulus* carbon concentration (open symbols indicate samples where *Hemiaulus* was not detected).

located sampling locations in the beginning of the cruise than at the end, suggesting that the bloom may have peaked
before the cruise started and thus was in decline during the cruise period.
419   Monthly mean surface Chl-a concentrations from remote sensing were used to investigate if the observed
differences in Chl-a and productivity between the summers were related to differences in the timing of the cruise as
opposed to differences in in community composition (Fig. S3). In many of the summers (2018, 2021, and 2022),
Chl-a in July was actually higher than in August, suggesting that the timing of the 2019 cruise (end of August
instead of end of July) was not a factor in explaining the anomalously high productivity observed in August, 2019. If
anything, the change in timing of the 2019 NES-LTER cruise would lead us to expect the Chl-a to be lower in
August than in July and thus the high Chl-a observed in August, 2019 is even more startling. Satellite data cannot be
used to confirm the presence or absence of *Hemiaulus*. However, IFCB data from NES broadscale NOAA EcoMon
surveys from summer 2013 to 2023, many of which occurred in August, always showed minimal presence of
*Hemiaulus*, suggesting the observed bloom in August 2019 was indeed extraordinary and not simply related to the
timing of the 2019 LTER cruise (Fig. S1).

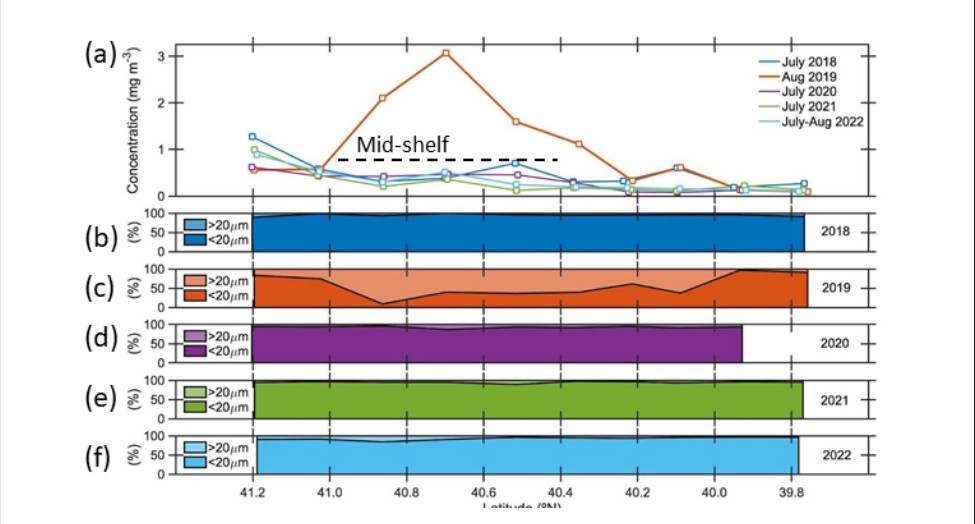

**Fig 4. (a)** Surface Chl-a concentration versus latitude for NES-LTER summer cruises 2018-2022. **(b-f)** Percentages of surface Chl-a associated with the > 20 μm phytoplankton (lighter shade) and < 20 μm phytoplankton (darker shade) versus latitude for each year. Note that the August 2019 cruise had the largest surface Chl-a concentrations and also the largest fractions associated with the > 20 μm fraction.

## 3.2 Physical properties

During the NES-LTER summer 2019 cruise, SST in the mid-shelf region was only slightly higher than during most of the other summer cruises (Table 2, Fig. 5). In contrast, SST in summer 2019 on the outer-shelf region in particular was substantially higher than on any other summer cruise (Fig. 6a). Notably, the 2019 cruise occurred later in the summer season (August) than the NES-LTER cruises in other years (July). Along the NES-LTER transect specifically, SST in July 2019 was lower than in August 2019 and was similar to other years. In general, monthly-averaged satellite SST data in the broader NES region usually showed lower SST values in July compared to August (2018, 2020-2022) (Fig. S3). Interestingly, however, in summer 2019, the monthly averaged satellite data actually showed higher SST in July, because of impingement of a Gulf Stream warm-core ring on the shelf edge (Zhang et al., 2023) and the subsequent onshore intrusion of the ring water in July 2019. The fact that monthly averaged satellite SST was higher in July than August but the local NES-LTER transect data had higher temperature in August than July suggests that the high SST observed during late August 2019 reflected an ephemeral event and not a mean condition during that month. Despite the occurrence of the NES-LTER summer 2019 cruise during a specific week of August and conditions that suggest an ephemeral event, for simplicity, we will refer to it as August 2019 in this paper. During the NES-LTER August 2019 cruise, surface salinity was lower than on 2018, 2021, and 2022 summer cruises, but similar to surface salinity during the July 2020 cruise (Fig. 5, Fig. 6b) and to salinities observed in July 2019 along the NES-LTER transect

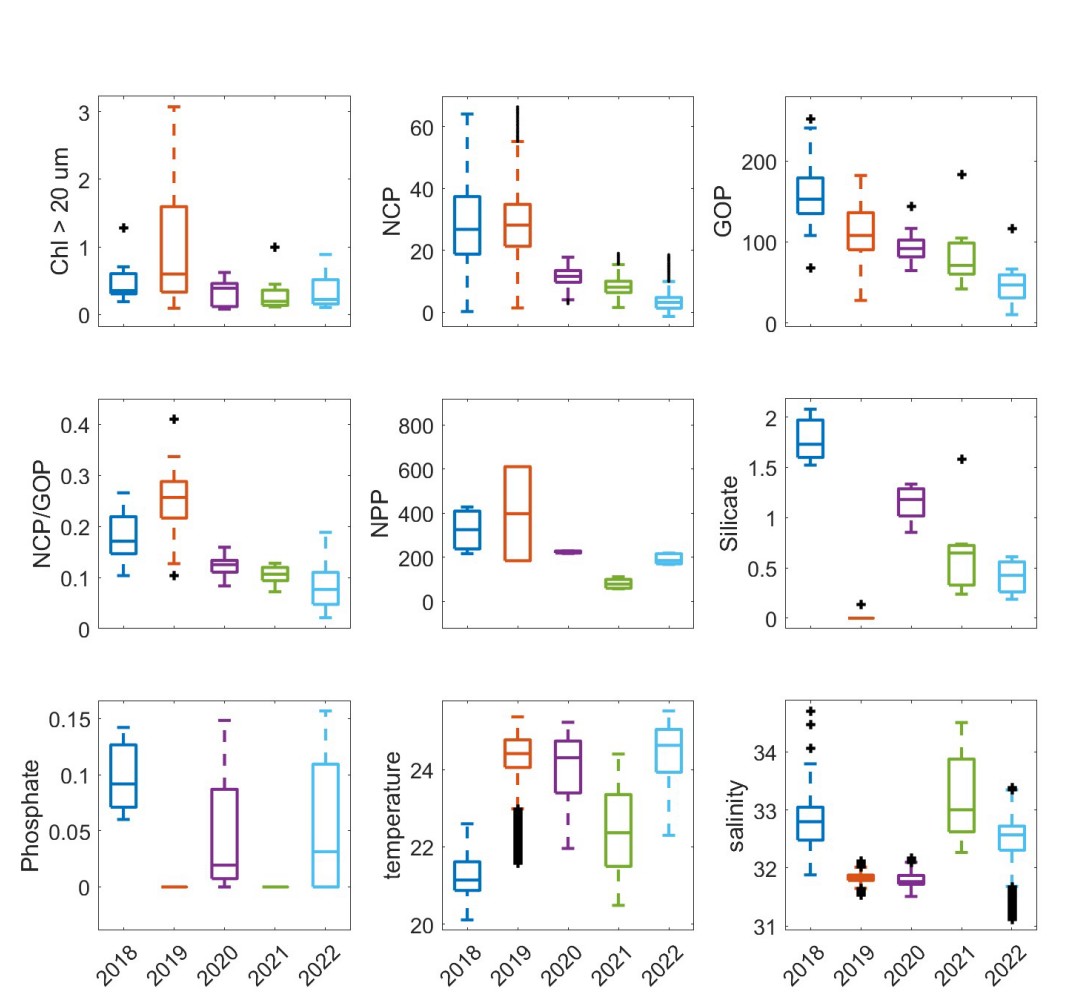

**Fig. 5.** Box plots of data in the summer, mid-shelf region for a) chlorophyll associated with cells > 20 μm in units of mg m⁻³, b) net community production (NCP) and c) gross oxygen production (GOP) both in units of mmol $O_2$ m⁻² d⁻¹, d) NCP/GOP (unitless) which is a measure of export efficiency, e) Net Primary Production (NPP) in units of mg C m⁻² d⁻¹, f) silicate and g) phosphate, both in units of μmol L⁻¹, h) sea surface temperature in degrees Celsius and i) salinity in psu. These plots show the differences in the plotted variables that occurred in August 2019 (orange box in each plot), a year when *Hemiaulus* carbon equaled 28.4 ug L⁻¹, compared to the data from the other summers, all of which had *Hemiaulus* carbon <0.02 μg L⁻¹.

**Table 2.** Averages, standard errors, and number of measurements (n) of surface mixed layer characteristics (productivity values integrated through the surface mixed layer; physical conditions, nutrients, and Chl-a concentrations from the surface) in the mid-shelf region (70.883° W, 40.437 °N - 40.980 °N, water depth 50 to 100 m) measured during NES-LTER summer cruises for each year. The last columns show the mean data for winter NES-LTER cruises 2018-2022 in the same mid-shelf region. The NPP average in 2018 (*) was calculated based on the phytoplankton growth rate since direct NPP measurements were not available for this year – see text for details.

| | 2018 | | | 2019 | | | 2020 | | | 2021 | | | 2022 | | | All winters | | |
|---|---|---|---|---|---|---|---|---|---|---|---|---|---|---|---|---|---|---|
| | **mean** | std | n | **mean** | std | n | **mean** | std | n | mean | std | n | **mean** | std | n | **mean** | std | N |
| NCP ($mmol\ O_2\ m^{-2}d^{-1}$) | **28.8** | 0.2 | 4816 | **28.8** | 0.2 | 5446 | **11.3** | 0.1 | 2895 | **8.4** | 0.05 | 6120 | **3.8** | 0.1 | 5763 | **5.1** | 0.1 | 29258 |
| GOP ($mmol\ O_2\ m^{-2}d^{-1}$) | **160** | 10 | 19 | **110** | 9 | 22 | **100** | 7 | 17 | **104** | 26 | 12 | **40** | 5 | 28 | **101** | 20 | 89 |
| NCP//GOP | **0.18** | 0.01 | 19 | **0.24** | 0.09 | 22 | **0.13** | 0.01 | 17 | **0.09** | 0.01 | 12 | **0.07** | 0.01 | 28 | **0.11** | 0.04 | 89 |
| NPP ($mg\ C\ m^{-2}d^{-1}$) | **\* 324** | 20 | \*4 | **398** | 213 | 2 | **225** | 4 | 3 | **81** | 10 | 6 | **191** | 10 | 6 | **464** | 63 | 11 |
| Phytoplankton growth rate ($d^{-1}$) | **1.12** | 0.11 | 4 | **0.2** | 0.17 | 2 | **0.8** | 0.10 | 2 | **0.83** | 0.64 | 2 | **0.31** | 0.1 | 2 | **0.3** | 0.04 | 18 |
| Micro-zoop. grazing ($d^{-1}$) | **0.19** | 0.03 | 4 | **0.17** | 0.04 | 2 | **0.22** | 0.09 | 2 | **0.51** | 0.25 | 2 | **0.11** | 0.05 | 2 | **0.24** | 0.05 | 18 |
| Temperature (°C) | **21.44** | 0.01 | 4816 | **24.29** | 0.01 | 5446 | **23.88** | 0.02 | 2895 | **22.61** | 0.02 | 6120 | **24.46** | 0.01 | 5763 | **6.131** | 0.007 | 29258 |
| Salinity (psu) | **32.71** | 0.01 | 4816 | **31.83** | 0.001 | 5446 | **31.8** | 0.002 | 2895 | **33.17** | 0.01 | 6120 | **32.53** | 0.01 | 5763 | **32.79** | 0.002 | 29258 |
| *Hemiaulus* Carbon ($\mu g\ L^{-1}$) | **0.002** | 0.002 | 93 | **28.4** | 1.3 | 102 | **0** | 0.00 | 65 | **0.001** | 0.001 | 113 | **0.003** | 0.003 | 104 | **0.018** | 0.006 | 562 |
| Chl-a ($mg\ m^{-3}$) | **0.43** | 0.10 | 8 | **1.97** | 0.59 | 8 | **0.4** | 0.06 | 8 | **0.22** | 0.07 | 8 | **0.32** | 0.10 | 8 | **2.17** | 0.16 | 40? |
| %Chl-a >20 μm | **0.02** | 0.006 | 8 | **1.37** | 0.43 | 8 | **0.03** | 0.01 | 8 | **0.01** | 0.004 | 8 | **0.03** | 0.02 | 8 | **1.56** | 0.14 | 38 |
| Silicate ($\mu mol\ L^{-1}$) | **1.9** | 0.3 | 8? | **0.27** | 0.5 | 8? | **1.4** | 0.5 | 8? | **0.75** | 0.4 | 8? | **0.42** | 0.18 | 8? | **1.7** | 0.5 | 40 |
| Phosphate ($\mu mol\ L^{-1}$) | **0.11** | 0.01 | 8? | **0.025** | 0.02 | 8? | **0.085** | 0.10 | 8? | **0** | 0.00 | 8? | **0.06** | 0.06 | 8? | **0.5** | 0.30 | 40 |

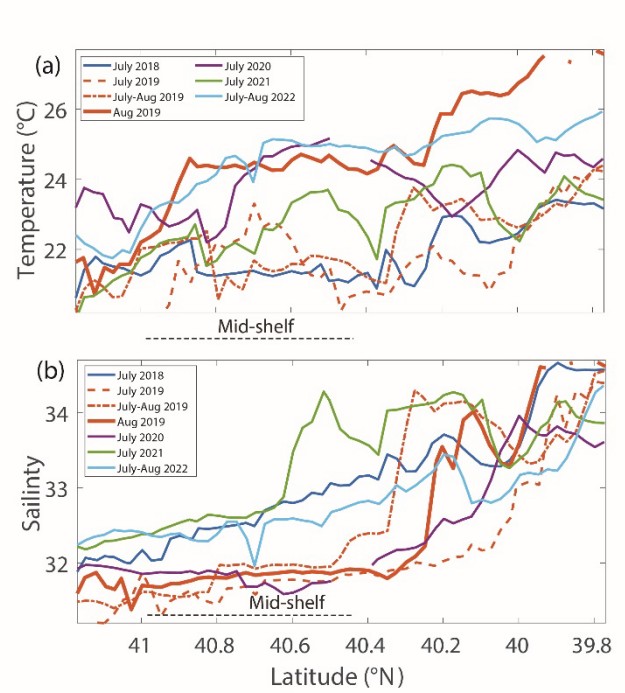

**Fig 6. (a)** Temperature and **(b)** salinity 5 m below the surface versus latitude for NES-LTER summer cruises (2018-2022) and the SPIROPA and OTZ summer 2019 cruises. For clarity, the values are averaged in 0.025 degree latitude bands when there were multiple occupations of the same region. The mid-shelf region is denoted by a dashed line.

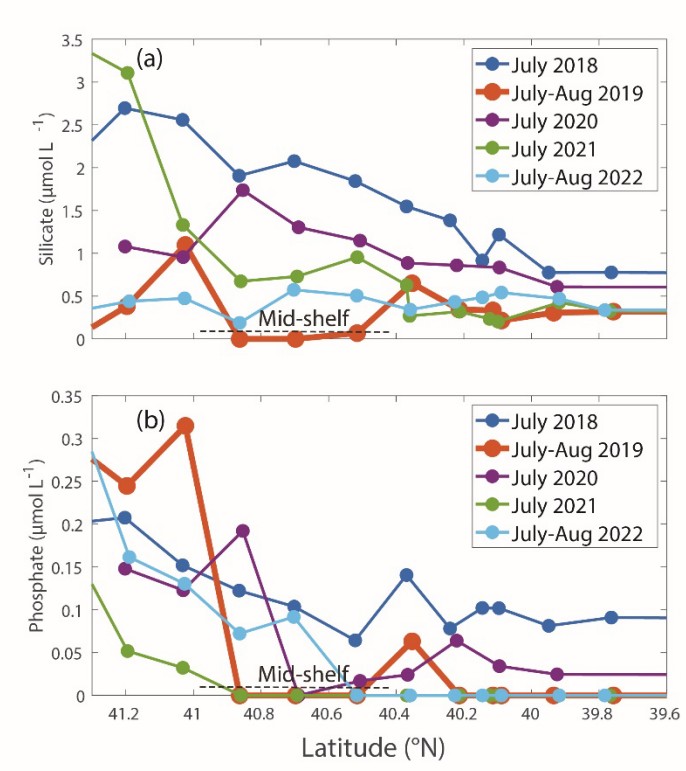

**Fig 7. (a)** Silicate and **(b)** phosphate concentrations (in µmol $L^{-1}$) ) in the upper 12 m of the water column for NES-LTER summer cruises (2018-2022). The mid-shelf region is denoted by a dashed line.

## 3.3 Nutrients

Nutrient concentrations differed between the August 2019 cruise and other summer cruises. Specifically, phosphate and silicate concentrations in surface waters were lower in August 2019 compared to most other summers (Table 2, Fig. 5, Fig. 7). In other summers, silicate decreased with distance from shore, but in 2019, silicate was depleted between 41ºN and 40.4 ºN (Fig. 7a) coincident with the location of the *Hemiaulus* bloom. Additionally, higher levels of silicate were found around depths of 50 m to 140 m in August 2019 than during other summer NES-LTER cruises (Fig. S4), which may be associated with diatoms that had sunk and were starting to be remineralized, releasing silicate back into the water column. Surface water phosphate concentrations in August 2019 were depleted south of 41º N (Fig. 7b). However, low concentrations of phosphate were also found in summer of 2021. Lastly, while nitrate plus nitrite were measured on the same samples as phosphate and silicate, nitrate + nitrite concentrations were close to the detection level in the surface samples for all summer cruises except a few stations in 2018 and thus are not shown here. Ammonium levels are not discussed because the samples were frozen at sea and thus may not be reliable; additionally, ammonium levels showed no clear relationship over the transect cruises.

## 3.4 Productivity and grazing rates

In August 2018 NCP was elevated in the mid-shelf waters, coincident with the location of the *Hemiaulus* bloom (Fig. 1). NCP peaked in the first half of the cruise and decreased during the second half, supporting the earlier supposition that the *Hemiaulus* bloom was likely in decline (Fig. S2). Additionally, the area of maximum NCP moved shoreward in the second half of the cruise. The high NCP was primarily constrained to the main longitude sampling line and usually did not extend, at least at those points in time, spatially off the main transect.

During August 2019, waters with high carbon concentrations of *Hemiaulus* showed higher rates of NCP (Fig. 8a), NCP/GOP (Fig. 8b), GOP (Fig. 8c), and NPP (Fig. 8d) compared to these rates at mid-shelf waters in most other years (Fig. 5). More specifically, the mid-shelf waters where *Hemiaulus* was present in Aug 2019 displayed NCP values approximately 2.5 to 9 times larger than in the same mid-shelf latitudes in summers of 2020-22 (Table 2). Furthermore, we

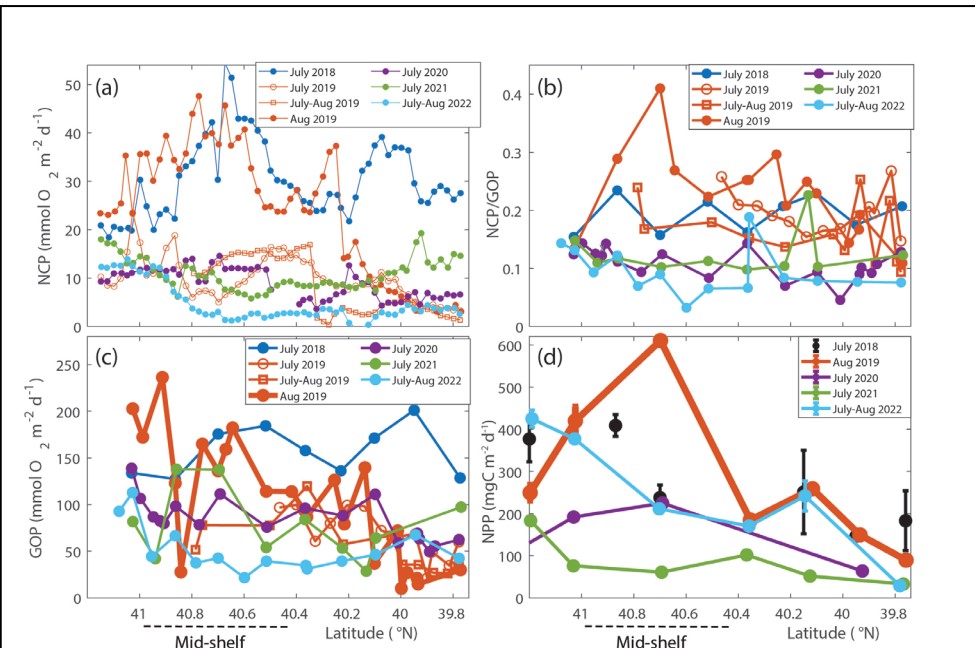

**Fig 8. (a)** NCP, **(b)** NCP/GOP, and **(c)** GOP rates integrated throughout the mixed layer for NES-LTER summer cruises (2018-2022) and SPIROPA and OTZ summer 2019 cruises. Values are averaged in 0.025 degree latitude bands to average multiple occupations of the same region. The same overall patterns are seen with and without the averaging within these latitude bands. **(d)** Average NPP values integrated to the bottom of the mixed layer for NES-LTER summer cruises (2019-2021) with error bars reflecting the standard deviation of triplicate surface water incubations. NPP values were not directly measured for summer 2018 but were instead estimated from phytoplankton growth rate in the grazing incubation experiments (black circles). The mid-shelf region is denoted by a dashed line.

observed a correlation between NCP and *Hemiaulus* carbon between Aug 21 and Aug 23 (Fig. 9; $R^2 = 0.54$, *p* <

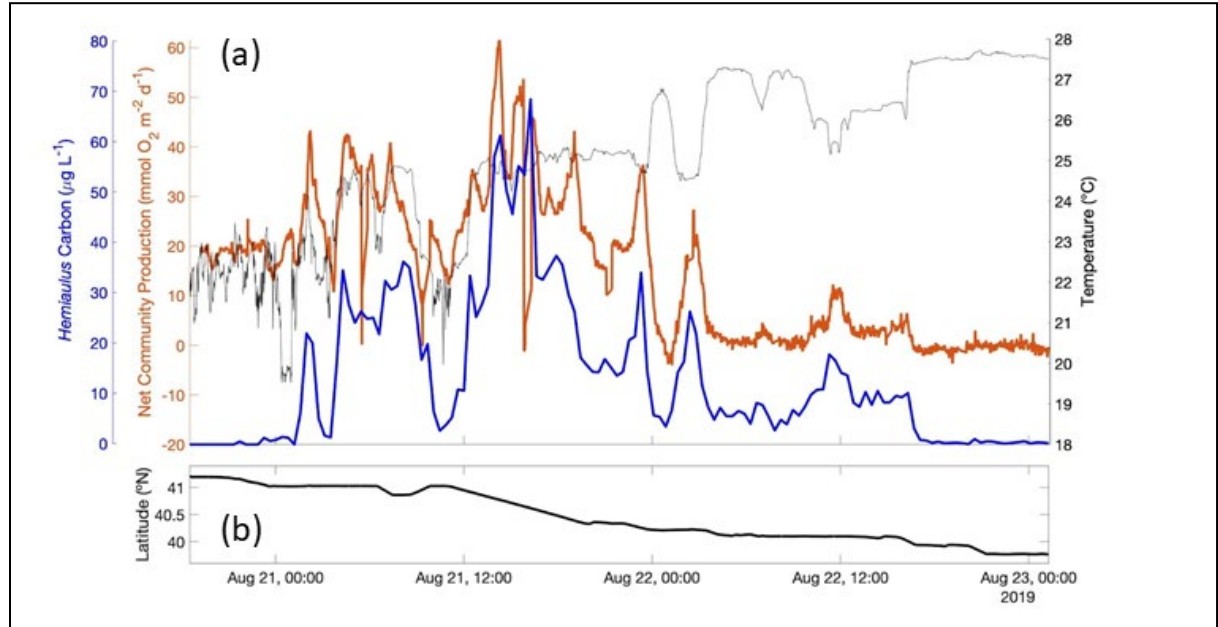

**Fig. 9. (a)** *Hemiaulus* carbon concentration (blue), as estimated from IFCB data, showing a strong positive correspondence with NCP (orange) ($R^2 = 0.54$, $p < 0.001$) and a weak negative correlation with SST (black) ($R^2 = 0.07$, $p = 0.002$) during one of the times during the August 2019 NES-LTER cruise that the ship was transiting the mid-shelf region, and **(b)** the latitude of the ship during the time period reflected in panel a.

0.001). The patchiness of the diatom bloom corresponded to the patchiness in NCP. Additionally, cooler shelf water
was associated with higher abundances of *Hemiaulus* than the warmer slope water (Fig. 9), suggesting a water mass
dependence on the location of the *Hemiaulus* bloom. Thus, the patchiness in the bloom and NCP is likely a result of
the ship crossing different water masses.
GOP rates were only slightly higher in summer 2019 than in other summers (Fig. 5 and 8c). In particular,
GOP rates were higher by a factor of 1.1 in waters with the *Hemiaulus* bloom in 2019 than during the summers of
2020-21; GOP rates  were much higher in August 2019 than in summer of 2022 by a factor of 2.75. Notably, NCP,
GOP, and NCP/GOP rates in summer 2018 were comparable to these rates  in August 2019 (discussed below in
section 4.1).
Additionally, within the region that corresponds directly with the location of the *Hemiaulus* bloom, NPP
rates in 2019 were ~1.5 - 2.5 times higher than NPP rates during other summer cruises. (Fig. 8d; Table 2). More
specifically, NPP at 40.7 °N was approximately double the NPP measured in 2020 and more than double the rate
measured in 2021. Furthermore, at 40.4 °N, NPP in 2019 was about 40% higher than in 2021 (no data for this station
in 2020) (Fig. 8d).
A larger difference between NCP in the various summers than between GOP in the summers suggests that
the increase in NCP in August 2019 was due to both increased photosynthesis and decreased community respiration.
In a rough approximation, we calculated autotrophic respiration and heterotrophic respiration to show that
autotrophic respiration was lower than average in August 2019 ($R_A = 308$ mg C m$^{-2}$d$^{-1}$ in August 2019 versus 496
mgC m$^{-2}$d$^{-1}$ average for the other summers). This approach also showed that heterotrophic respiration was higher
than average in August 2019 (431 mgC m$^{-2}$d$^{-1}$ in August 2019 versus 247 mg C m$^{-2}$d$^{-1}$ average for the other
summers). Note this estimation is highly uncertain due to the different time and spatial scales associated with the gas
tracers used to quantify NCP and GOP and the incubation techniques used for NPP.
Since the summer 2019 NES-LTER cruise occurred in the middle of August rather than in mid to late July
as was typical for most other summers, the physical conditions were inherently different in 2019. We compared
NCP and GOP data (NPP not available) from two earlier cruises in summer 2019 (cruise details in Table 1) whose
cruise track in the mid-shelf region overlapped with those of the LTER cruise, i.e. followed the same longitude
70.883°W. These cruises occurred before the *Hemiaulus* bloom and while their IFCB records showed a detection of
*Hemiaulus*, the abundance of this diatom was very low (< 1 µgC L$^{-1}$). These two July 2019 cruises had much lower
NCP rates compared to August 2019, and specifically had rates similar to those observed in summer 2020-22 NES-

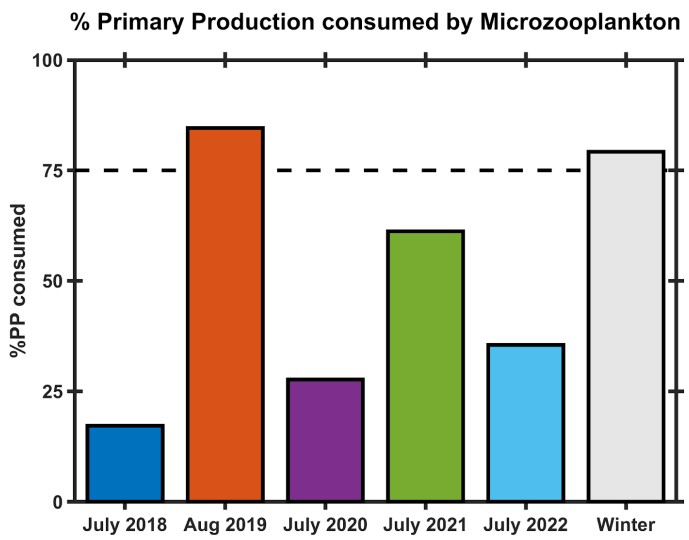

**Fig 10.** Percentage of primary production (%PP) consumed by microzooplankton in surface waters in the mid-shelf region during NES-LTER cruises for each summer and the overall average from NES-LTER winter cruises 2018-2022. %PP consumed by microzooplankton is calculated as the ratio of microzooplankton grazing rate ($d^{-1}$) to phytoplankton growth rate ($d^{-1}$).

LTER transect cruises (Fig. 8a & b). Together, these data suggest that higher production rates in 2019 were uniquely
tied to the presence of *Hemiaulus* rather than representing deviations in timing or environmental conditions.
The ratio of protistan grazing to phytoplankton growth rates provides an estimate of the percent of primary
production (%PP) consumed by microzooplankton (Fig. 10). In contrast to typical summer conditions (> 0.6 $d^{-1}$,
Table 2), during August 2019, phytoplankton growth rates during the *Hemiaulus* bloom were low (< 0.2 $d^{-1}$, Table
2) compared to other summers, with most of the primary production consumed by microzooplankton (%PP = 84%,
Table 2). Notably, these low phytoplankton growth rates are in the same range as other diatoms with Richelia
symbionts, namely 0.3 $d^{-1}$ for *Rhizosolenia-Richelia* cultured at a similar temperature (Villareal, 1990) . Thus in
August 2019, phytoplankton growth and microzooplankton grazing were well coupled (though only in the part of the
transect where *Hemiaulus* bloomed), like typical winter conditions, when the phytoplankton community structure is
dominated by large cells, instead of the decoupling typically observed in most summer conditions dominated by
picoplankton (Marrec et al., 2021). We note that coupling between phytoplankton growth and microzooplankton
grazing was occasionally observed during other summer cruises, but mostly in inner-shelf waters (except one mid-
shelf station in July 2021). Overall, most of the primary production during the *Hemiaulus* bloom was grazed by
microzooplankton indicating high trophic transfer efficiency from phytoplankton to microzooplankton.
**4 Discussion**
**4.1 Change in community composition altering biological rates**
A bloom of *Hemiaulus* has not been observed on any previous NES-LTER cruise and to our knowledge,
has not been reported in the broader NES region before. Additionally, only in August 2019, compared to summers
2018 and 2020-22, most of the Chl-a was associated with the > 20 μm size fraction. Thus, the presence of the diatom
bloom found in August 2019 was a major change in phytoplankton composition observed in this region of the NES
that likely led to the observed large changes in productivity rates and coupling between phytoplankton growth and
grazing.
It is likely that the nitrogen-fixing symbionts in *Hemiaulus* allowed the diatom to thrive in the stratified,
low nutrient surface waters of the summer shelf. This is supported by phosphate draw-down to levels below
detection in August 2019; the nitrogen-fixing symbionts in the *Hemiaulus* likely made phosphate a limiting factor
for growth (Tang et al., 2020) in August 2019 whereas nitrate limitation is typical for NES summer conditions.
While the summer of 2021 also had very low phosphate, summer 2021 was different in that it also had low
productivity rates and more typical levels of silicate, suggesting the low phosphate occurred for fundamentally
different reasons in 2019 and 2021.
Silicate is especially important for diatoms because it is required for formation of their cell frustules.
Moreover, previous studies show that the availability of dissolved silica seems to be an important control for many
diatom-diazotroph blooms by affecting the growth rate and size of the diatom's frustules (Kemp and Villareal, 2013;
Spitzer, 2015). The observed depletion of silicate and phosphate in surface waters during the August 2019 cruise
suggests that, at the time of the cruise, the *Hemiaulus* bloom may have been in decline. The low phytoplankton
growth rates of 0.2 day$^{-1}$ support the idea that the bloom had peaked, particularly given the fact that with sufficient
phosphate, silicate and light, the *Hemiaulus* DDA can achieve growth rates of 0.7-0.9 d$^{-1}$ in laboratory cultures (Pyle
et al., 2020). Low growth rates also could be attributed to the inverse relationship between phytoplankton cell size
and growth rate. The *Hemiaulus* population could have been limited by phosphate, silicate, or both. The higher
levels of silicate observed at depth in August 2019 are likely due to *Hemiaulus* sinking out of the euphotic zone and
frustule remineralization at depth, which would release the silicate–and other nutrients–back into the water (Twining
et al., 2014).
The strong coherence between the high spatial resolution data on *Hemiaulus* carbon concentrations and
NCP (Fig. 9), as well as the other data presented here and a clear potential mechanism, strongly support the idea that
the high productivity rates observed in August 2019 are directly due to the presence of *Hemiaulus*. In particular, the
high NCP rates observed during the August 2019 NES-LTER cruise and their overlap with the location of the
diatom bloom, suggest a high export ecosystem developed due to *Hemiaulus*' influence on productivity and
biological rates. Here, we define export as a flux away from the local biological production compartment, which can
include losses of carbon (or oxygen) to depth or transfer to higher trophic levels. While the *Hemiaulus* bloom
slightly increased total photosynthesis, as seen from the GOP rates, the bloom presence affected NCP, and thus by
extension, export production to a higher degree, potentially due to the large size of *Hemiaulus* cells and chains. The
NCP/GOP ratio in August 2019 was double the ratio observed in the summers 2020-22 (Table 2, Fig. 5). Other
studies have shown links between variations in NCP/GOP and changes in planktonic community composition
(Palevsky et al., 2016). Bigger phytoplankton cells sink faster than small ones, making them less likely to be grazed
before sinking out of the euphotic zone, allowing for a higher export efficiency. Additionally, a higher trophic
transfer efficiency (see next paragraph) would also lead a larger NCP/GOP ratio. Hence, the NES-LTER summer
2019 cruise appears to have represented a high carbon export efficiency system.
Not only did NCP and GOP rates change because of the *Hemiaulus* bloom, but so did NPP, phytoplankton
growth rates, chl-a concentrations, and the trophic transfer efficiency within the planktonic food web. The presence
of *Hemiaulus* in the mid-shelf region likely led to the observed higher NPP rates during August 2019 compared to
all other observed summers in the mid-shelf region of the NES (Fig. 5 and 8). High NPP rates associated with
diatom blooms have been observed in other systems such as on the Eastern Bering Shelf (Lomas et al., 2012) and in
the Gulf of California (Puigcorbe et al., 2015), including during blooms of, diatom-diazotroph associations such as
*Hemiaulus-Richelia* (Gaysina et al., 2019). For example, Tang et al. (2020) reported a high contribution of nitrogen
fixation to NPP off the coast of New Jersey during their 2015-2016 survey in the Western North Atlantic. Even
though high NPP was associated with the location of the *Hemiaulus* bloom in our study, phytoplankton growth rates
were low (< 0.2 d$^{-1}$). This decoupling between NPP and growth was likely due to the order of magnitude higher chl-
a concentrations observed during August 2019 (1.37 µg L$^{-1}$) compared to other summers (0.01 - 0.03 µg L$^{-1}$; Table
2) since NPP is roughly the product of phytoplankton growth and biomass (Marchetti et al. 2009). Thus, although
growth rate was low, biomass was so high that NPP was also high. Furthermore, most of the primary production was
directly consumed by microzooplankton, which we have not observed during any other summer NES-LTER cruise,
suggesting the presence of *Hemiaulus* led to more efficient trophic transfer during August 2019. While conditions
with high NCP (i.e. low community respiration) and high grazing pressure as observed in August 2019 may seem
counterintuitive, they are not contradictory since grazing cannot be equated with respiration. First, much of
respiration is bacterial and therefore not reflected by the grazing rates (Robinson and Williams, 2005). Second, it has
been observed that after starvation, protozoan grazers increase their organic matter production by accumulating
lipids and increasing their cell size (Anderson and Menden-Deuer, 2017; Morison et al., 2020). Thus, high grazing
could suggest a buildup of organic matter through secondary production, which is consistent with the higher than
average microzooplankton biomass and would be reflected as large NCP. Third, microzooplankton can produce
fecal pellets (Buck and Newton, 1995), which removes carbon from the system without respiration and leads to high
NCP. The dominant presence, and slow growth, of large *Hemiaulus* cells within  the phytoplankton community was
likely a main factor promoting the higher trophic transfer efficiency from phytoplankton to microzooplankton, as is
typical during winter (Marrec et al., 2021).
Interestingly, NCP and GOP values in summer 2018 were similar to those in August 2019 (Table 2, Fig. 5
and 8) and also much higher than during subsequent summers (2020-2022), in spite of no *Hemiaulus* being present
in summer 2018. Additionally, the ratio of NCP/GOP in summer of 2018 was also significantly larger than in 2020-
22 (Fig. 5 and 8 ). Remote sensing shows an elevated Chl-a patch (less concentrated than the patch in August 2019)
in summer 2018 west of the transect that could be the driving factor behind the high NCP and GOP values (Fig.
S3). The summer of 2018 was dominated by small phytoplankton similar to observations in summers of 2020 and
2021, although the summer of 2018 had a particularly high concentration of dinoflagellates over parts of the shelf.
The summer 2018 data did not show an increase in trophic transfer efficiency due to coupled microzooplankton
grazing and phytoplankton growth. High NCP rates in summer 2018 could be due to a variety of environmental
(biotic and abiotic) factors that were different from other cruises. For example, in the summer of 2018, saline waters
from offshore intruded much farther inshore than during most of the other summers and these high-salinity mid-
shelf waters were particularly productive (Mehta, 2022). Additionally, correspondence was seen between NCP and
dinoflagellate biomass in summer 2018, although this correlation was not as significant as that between *Hemiaulus*
and NCP in 2019 (Aldrett, 2021). Thus, this study shows that a change in community composition, such as the
*Hemiaulus* bloom in August 2019, can dramatically change the productivity rates of the ecosystem even though a
very different  phytoplankton community structure can sometimes lead to similarly high productivity.
**4.2 Aggregate vs Compositional Variability**
The changes in community composition, productivity rates, and chlorophyll in August 2019 compared to the other
summers shed interesting light on the question of synchrony or compensation between aggregate and compositional
variability at the NES-LTER site (Micheli et al., 1999; Shoemaker et al., 2022). The resilience of an ecosystem may
be related to the compensation or synchrony between different types of variability (Lindegren et al., 2016). During
August 2019, the phytoplankton composition in the NES changed dramatically due to the bloom of the diatom
*Hemiaulus*. This change was associated with increases in Chl-a, higher productivity rates, tighter coupling between
microzooplankton grazing and phytoplankton growth, and increases in export efficiency. These latter terms are all
metrics of aggregate properties and thus this bloom event exhibited high compositional and high aggregate
variability compared to the ecosystem in July of 2020-22. Thus, during this event, a metric associated with
compositional variability (e.g., the change in phytoplankton community composition) was synchronous with metrics
associated with aggregate variability. However, when NCP rates are compared from the summer 2018 to summer
2019, the composition is still quite different (*Hemiaulus* in 2019 compared to mostly small phytoplankton in 2018)
and thus there is still large compositional variability  but the aggregate properties in terms of NCP is similar in both
years, showing that sometimes compensation occurs in which the community composition changes but the aggregate
productivity does not. This concurrent investigation of plankton community composition and production rates within
a well-studied ecosystem highlights how shifts in community size distribution can greatly affect productivity.
However, it also shows that multiple factors change from year to year, leading to different effects.
**4.3 Origin of Bloom**
The *Hemiaulus* bloom was likely more widespread than what was observed in the NES-LTER 2019 summer cruise.
For example, satellite imagery from August 11 shows a filament of warm, high Chl-a waters oriented southwest-
northeast and ending in the region where *Hemiaulus* was abundant (Fig. 11a & b); the advective continuity of the
filament with the *Hemiaulus* patch suggests the filament may have had high *Hemiaulus* as well. Direct support for a
widespread bloom comes from IFCB data collected on the NOAA EcoMon Cruise (GU1902) that occurred at a
similar time as the August 2019 NES-LTER transect cruise. The IFCB data shows that *Hemiaulus* was present both
farther east as well as to the southwest of where it was observed on the LTER transect cruise and that some of the
points in the high chlorophyll filament observed from satellite chlorophyll contained *Hemiaulus* (Fig. 11c).
Backward particle trajectory analysis based on HF radar-measured sea surface velocities show that the
water with high *Hemiaulus* biomass during the August 2019 transect cruise could have been advected from the
inner-shelf around Narragansett Bay and Georges Bank rather than from the mid-shelf further south (Fig. S6). In
particular, coastal upwelling probably brought the inner-shelf water into the mid-shelf transect area where it was
observed to have high *Hemiaulus*. The salinity of the water with the high *Hemiaulus* biomass is consistent with the
water having originated from the shelf. The water with high amounts of *Hemiaulus* carbon was associated with
salinity ranging from 31.6 to 34 psu and temperatures of 22° C to 27° C (Fig. 12a & b). T-S plots of data from other
years (Fig. 12c) suggest that
several other summers also had
similarly warm, low salinity
water (in particular July of 2020
and especially 2022) but
interestingly *Hemiaulus* were
not observed on those cruises.
Although multiple lines
of evidence suggest that the
water containing the high
biomass of *Hemiaulus* initially
originated from the inner-shelf,
*Hemiaulus* is typically found in
warm, low nutrient water –
characteristics that are not
present on the inner-shelf, where
water is instead colder and
nutrient-rich. In this case, the
inner-shelf water warmed as it
was transported offshore and
thus it reached temperatures
warm enough for *Hemiaulus* to
thrive by the time it reached the
mid-shelf (the timing of
warming is not known). But
how did this inner-shelf water
acquire *Hemiaulus* as it was
transported offshore in August
2019? One possibility is that it
was seeded by the warmer low
nutrient surface slope and ring
waters; in particular, these slope
and ring waters were observed
earlier in the summer of 2019 to
have a small population of
*Hemiaulus* that could have
served as a seed population
(Oliver et al., 2021). However,
there is no evidence of surface
transport from slope-water to the
*Hemiaulus* patch. Another
possibility is that *Hemiaulus*
were already present in the
deeper coastal water and then
thrived as the deep water was
mixed upward, warmed and
reached the higher light surface
waters. However, the vertical

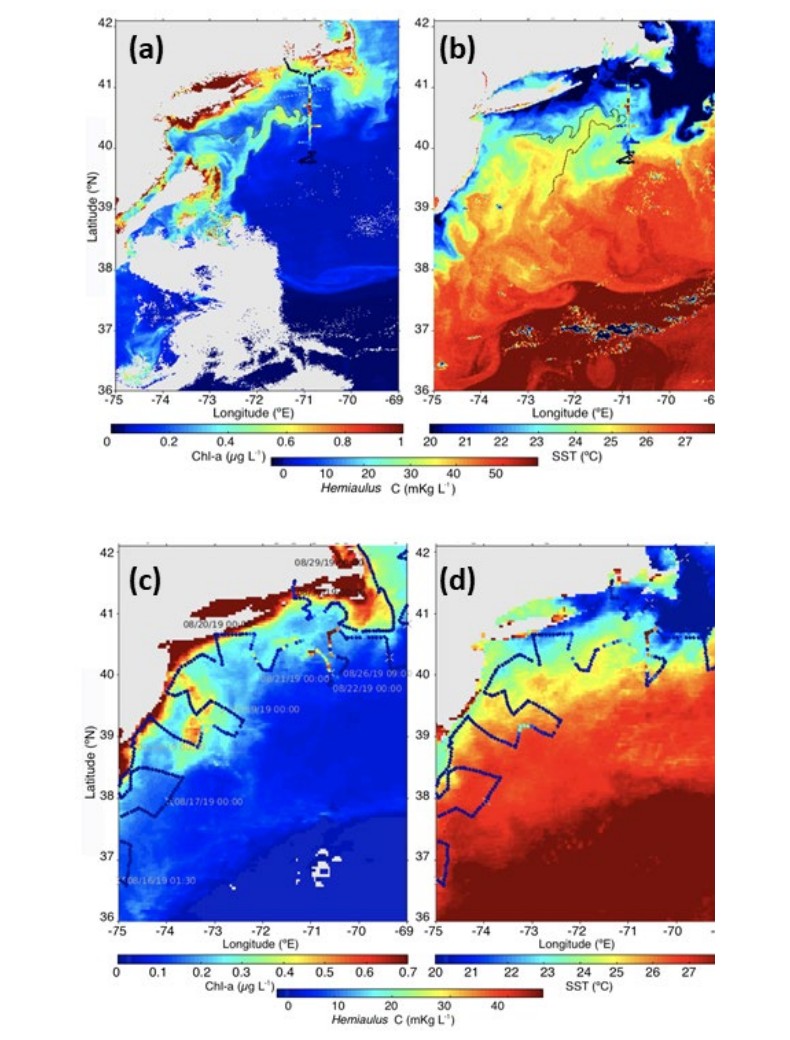

**Fig 11.** Snapshots of satellite-measured **(a)** Chl-a and **(b)** SST on 11 August 2019. The 0.5 µg L⁻¹ chlorophyll contour is plotted as a solid line. *Hemiaulus* carbon from underway surface samples during the NES-LTER Aug 2019 cruise is overlaid with colored dots. The monthly composite **(c)** Chl-a and **(d)** SST for August 2019. *Hemiaulus* carbon from underway surface samples during EcoMon cruise GU1902 is overlaid with colored dots. Daily ship positions are indicated in the left panel.

distributions of *Hemiaulus* (Fig. 3) do not support this hypothesis, since a deeper population was not
observed. Finally, the modeled backward particle trajectories suggesting an inner-shelf origin may be inaccurate as
shelf water circulation is complex, as seen by the conflicting origins of the *Hemiaulus* water to both the east and
west of the transect and an inconsistency between conclusions from the particle trajectory analysis with the high
Chl-a, high temperature filament observed in the Satellite imagery (Fig S6). Thus, the reason *Hemiaulus* bloomed in
2019, and not in other years, remains a topic for future research and continued speculation. Future years of the NES-
LTER program may shed light on the variable effects of disturbances, such as this *Hemiaulus* bloom, as more
factors that lead to high or low export in summer are determined and explored.

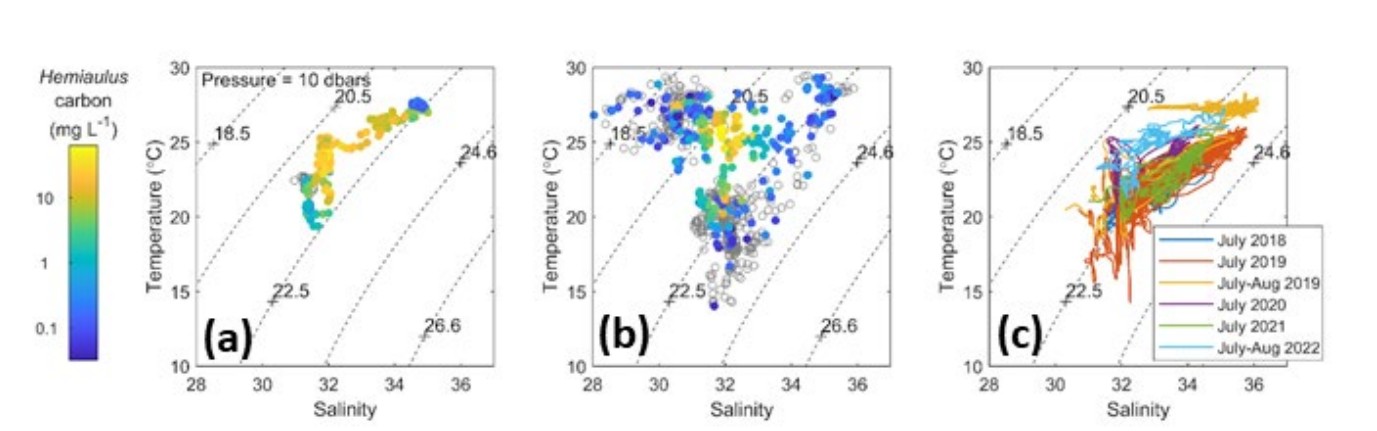

**Fig 12.** T-S plots, colored according to *Hemiaulus* carbon, suggesting that the highest *Hemiaulus* abundances were associated with a "sweet spot" in T-S space that was found during **(a)** the 2019 summer NES-LTER cruise in August and **(b)** the EcoMon August 2019 cruise. Empty circles represent locations where *Hemiaulus* was not detected. **(c)** TS plots from other years, colored according to cruise, show that a few of the other summer cruises from 2018 to 2022 have the same sweet spot in physical conditions even though they did not have detectable *Hemiaulus*.


## 5 Conclusions

An unusual bloom of the diatom genus *Hemiaulus* with nitrogen-fixing symbionts in the mid-shelf region of the Northeast U.S. shelf in August 2019 was observed concomitant with increases in NCP, GOP, NPP, higher export efficiency, and higher trophic transfer efficiency from phytoplankton to microzooplankton. Very tight coupling observed between kilometer-scale changes in NCP and the carbon biomass of *Hemiaulus* showed a substantial effect from the *Hemiaulus* bloom on important biogeochemical rates and stocks of the Northeast U.S. Shelf. While the source of the *Hemiaulus* on the inner-shelf remains unknown, the bloom was associated with warmer temperatures than usually observed on the shelf which may have been an important factor that facilitated the bloom when it was transported from the inner-shelf.

The *Hemiaulus* bloom, which was observed at a time when there were warmer sea surface temperatures especially in the outer-shelf region, was intriguing in that it led to unusually high productivity rates, increases in Chl-a concentrations, and tighter food-web coupling. While the warm SST may have contributed to the *Hemiaulus* bloom, the summer cruise of 2022 showed nearly as high water temperature as 2019 in the outer-shelf and the summers of both 2020 and 2022 had similarly high water temperatures as 2019 in the mid-shelf region. However, summers 2020 and 2022 had relatively low (i.e., average summer) productivity rates and Chl-a. So, summers 2020 and 2022 had fairly similar physical conditions to that of 2019, but no significant bloom was observed, and no high-carbon export system was present. Thus, higher temperatures are not enough to explain higher productivity rates, a shift in community composition is also necessary. A mixture of the right physical conditions and community composition, like this special case of 2019, are needed for a high-carbon export system to be supported on the mid-shelf during summer.

With climate change, the oceans are warming at a rapid rate, and are likely moving towards warmer more stratified conditions (e.g., lower nitrate stock in surface waters) (Li et al., 2020) which may lead to less productivity and thus lower export efficiencies. However, these conditions may also lead to unusual phytoplankton composition as species distribution shifts. The results presented here show that unusual events can lead to large locally and episodically enhanced productivity and export; despite a commonly nitrate-limited ecosystem during the summer season, an intense phytoplankton bloom in summer occurred due to a symbiotic diatom-diazotroph relationship. These observations lead to further questions about how the NES ecosystem is responding to the effects of climate change such as enhanced stratification. Monitoring future disturbances and their effects will provide new insights

into relationships, mechanisms, and patterns of composition and productivity that may be only occasionally
occurring now but are likely more prevalent in the future.

## 6 Data Availability

All in situ data are available at the EDI data repository. In particular, the raw gas tracer data used for calculating
NCP and GOP is available at
https://portal.edirepository.org/nis/mapbrowse?packageid=knb-lter-nes.6.2. The calculated rates of NCP data is
accessible at https://portal.edirepository.org/nis/mapbrowse?packageid=knb-lter-nes.7.2
and  https://portal.edirepository.org/nis/mapbrowse?packageid=knb-lter-nes.15.2. NPP data is available at
https://portal.edirepository.org/nis/metadataviewer?packageid=knb-lter-nes.16.4. Grazing rate data is available at
https://portal.edirepository.org/nis/mapbrowse?packageid=knb-lter-nes.5.1. Chlorophyll data is available at
https://portal.edirepository.org/nis/mapbrowse?packageid=knb-lter-nes.8.1. IFCB data is available at
https://portal.edirepository.org/nis/mapbrowse?packageid=knb-lter-nes.9.1 and on the IFCB dashboard
at  https://ifcb-data.whoi.edu/timeline?dataset=NESLTER_transect and https://ifcb-
data.whoi.edu/timeline?dataset=NESLTER_broadscale. .
The MODIS SST and chlorophyll snapshot data were produced by NASA Goddard Space Flight Center, Ocean
Ecology Laboratory, Ocean Biology Processing Group, and the data are publicly available at
https://oceancolor.gsfc.nasa.gov/. The 8-day composite data were retrieved from the public-accessible University of
Delaware ERDDAP server (https://basin.ceoe.udel.edu//erddap/index.html) maintained by the Ocean Exploration,
Remote Sensing and Biogeography Laboratory led by Dr. Matthew Oliver at University of Delaware. The HF radar-
measured sea surface velocity data in July-August 2019 was obtained from the public-accessible Rutgers University
Center for Ocean Observing Leadership  ERDDAP server (http://hfr.marine.rutgers.edu/erddap/griddap/).

## Author Contribution

SAC, RHRS, ZOS, and DA measured and calculated rates of productivity from gas tracers. SMD and PM measured
grazing rates. TAR and DNF measured and calculated rates of net primary productivity from bottle incubations.
HMS, ETC and EEP imaged and quantified phytoplankton abundances. DJM and WGZ analyzed remote sensing
data. Everyone participated in study design. SAC and RHRS prepared the manuscript with contributions from all co-
authors.

## Competing Interests

The authors declare that they have no conflict of interest.

## Acknowledgements

This work was funded by the National Science Foundation (LTER-1655686, OCE-1657489, OCE-1657803, OCE-
2227425) and the Simons Foundation (561126 to HMS). S. A. Castillo Cieza was supported by the Clara Boothe
Luce Fellowship program at Wellesley College. We are thankful for the scientific input, discussions and help from
the entire NES-LTER science team. We are grateful to the Captain and crew of the *R/V Endeavor.* We thank Harvey
Walsh, Jerome Prezioso, Audy Peoples and Tamara Holzwarth-Davis for their cooperation and enthusiasm for IFCB
operations on NOAA survey cruises. We recognize the contributions of Kevin Cahill (WHOI), who ran some of the
samples for triple oxygen isotope measurement, Elizabeth Lambert (Wellesley College) and Helene Alt (Wellesley
College) who helped collect some of the EIMS data, and Danielle Aldrett (Wellesley College) for doing some initial
analysis on connections between the IFCB and NCP data. We thank NES-LTER data manager Stace Beaulieu and
Kate Morkeski (WHOI) for their help in data management. We thank URI-GSO undergraduate and graduate
students and postdocs who helped collect samples and conduct experiments to obtain chl-*a* concentrations, and
phytoplankton growth and microzooplankton grazing rates. We thank Sam Setta for pointing out *Richelia* in IFCB
images during the 2019 summer cruise. DJM gratefully acknowledges NSF support of the SPIROPA program, and
technical assistance by Olga Kosnyrev in satellite data analysis and visualization.

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
