# Peer review of "Unusual *Hemiaulus* Bloom Influences Ocean Productivity in Northeast U.S. Shelf Waters"

_Biogeosciences, 2023_

## Author Comment (AC1)

Dear Editor,

We are thankful for the thoughtful reviews. Below we respond to Reviewer 1 in detail. Additionally, because of the suggestions of Reviewer 2, we have added one figure to the main manuscript and 2 figures to the supplemental material. In case Reviewer 1 would like to see these new figures, we are including them at the end of this reply.

In the response below,
Reviewer 1 text is in blue
Our response is in black.
*Direct quotes from the revised manuscript are in black italics.*

Sincerely,
Rachel Stanley (on behalf of all the authors)

Reviewer 1: I think this is an excellent paper, and the comments I have are so minor that I do not need to review it again.

The work is very well executed, and the authors make a strong case for the occurrence of a relatively large *Hemiaulus* bloom on the NES in 2019. The text itself is well written. I especially liked how the authors clearly presented their definitions of GOP, NPP, NCP and export efficiency in the introduction. Use of data from other years, to put the 2019 'anomaly' in context, was a great strength, as were the backward particle trajectory simulations

We gratefully acknowledge the praise of the reviewer.

Minor edits that I recommend for clarity:

- In the Introduction, line 86, large phytoplankton (LP) cells are defined as those > 10 um, but then a 20 um filter is used to define LP in the actual study. I suggest changing the intro to read something like "LP, defined as > 20 um in this study".

Thank you for pointing that out. We used the 20 μm data because we have it for every time and space point whereas we only have 10 μm for some of the samples The conclusions did not change whether we used the more limited 10 μm or the extensive 20 μm data but of course it is important for the text of the paper to represent what we did. Since this part (lines 83-89) of the introduction is about the general ecosystem rather than this particular study, we omitted the size definitions from the sentence the reviewer mentioned and instead just say large and small since different studies define those terms differently. In section 2.6 where we talk about size fractionation, we have added a sentence as follows:
*"Note that in this study large phytoplankton are defined as > 20 μm."*

-line 117 - why is the underway system giving only km-scale resolution? Do you mean that the samples were collected AT km-scale resolution..or every km?

The underway NCP data is limited by the time interval of the mass spectrometer measurement, which is every 30 seconds. Additionally, there is gas equilibration inside the liquicel cartridge that practically means each datapoint represents about 2 to 5 minutes. The ship traveled at 5 to 10 knots typically (the slower speeds were when we were towing a video plankton recorder) so this translates to 0.3 to 1.5 km spacing. The phytoplankton data were only collected every 20 minutes so the spacing for that data is 3 to 6 km depending on the speed. Our use of km scale resolution was being intentionally general to encompass the different types of measurements described in that sentence. But we have rephrased the sentence to be more clear. The salinity and temperature data that we used were one minute averages so represent data at 0.14 km spacing.

*"Some data during that event, such as surface seawater temperature (SST), salinity (SSS), NCP rates, and phytoplankton composition were collected continuously from the underway system (every 0.1 to 6 km depending on the measurement type and ship speed), while other parameters (e.g., NPP, grazing rates, Chl-a, nutrients) were measured discretely at the NES-LTER stations"*

-line 138 - Have previous studies proven that diaphragm pumps are less damaging? (If so, I suggest adding a citation). When making any physiological measurement I usually avoid pumps completely and use water collected by Niskin or other bottle.

The images from the IFCB (available at https://ifcb-data.whoi.edu/timeline?dataset=NESLTER_transect) show that many phytoplankton are easily identified and thus not very much damaged when they go through the diaphragm pump. In response the reviewer's suggestion, we have added a citation:

Cetinić et al. 2016. Characterizing the phytoplankton soup: pump and plumbing effects on the particle assemblage in underway optical seawater systems. Optics Express. DOI: 10.1364/OE.24.020703

-Some justification should be given for the choices of C:Chl and PQ values. C:Chl, especially, can be highly variable.

We agree that C:Chl ratios are highly variable, especially in coastal waters and have added in a reference to that effect. However, in this study, the C:Chl ratio is only used indirectly when calculating rates of NPP for 2018 since direct NPP measurements were not available during this summer. In particular, we first used the C:Chl ratio to convert phytoplankton growth rates from all years into an estimate of primary production. We then compared the PP calculated this way to the actual NPP measured for 2019-2022 and then determined the linear regression. We next adjusted our calculated rates for 2018 based on this linear relationship. Thus as long as we use the same C:Chl ratio for the entire analysis, the results are insensitive to C:Chl – the relationship between PP and NPP would change to compensate for a different C:Chl ratio. We have described this in more depth in the paper.

*"This calculated productivity from chlorophyll was then converted into NPP based on a linear relationship determined between Chl-calculated productivity and measured NPP during the summers when NPP rates were measured directly. While C:Chl ratios in coastal systems are*

*highly variable seasonally (Jakobsen and Markager, 2016), we used the same C:Chl ratio when calculating the linear relationship and when scaling up NPP, and thus the estimated NPP rates are insensitive to the choice of C:Chl ratio.  C:Chl ratios were not used for  NPP rate calculation for any other year. "*

-What is the average cell size for *Hemiaulus*? I know they are big, but how big? Please add scale bars to Figure 2.

Scale bars have now been added to Figure 2. Individual *Hemiaulus*  are typically slightly smaller than 50 μm and the chains collected were up to a few hundred μm- the new scale bars allow readers to judge this for themselves.

-Line 588 - growth rates for *Hemiaulus* from the dilution experiments are very similar to values for large (non-symbiont containing) *Rhizosolenia* species...suggest citing some of T. Villareal or T. Richardson's work.

We thank the reviewer for suggesting those references. Indeed, the growth rates we find are nearly the same as those reported in Villareal 1990 in which cultures were grown at the same temperature as was observed during our cruise. Note the rates reported in Villarreal 1990 have to be multiplied by the natural log of 2  to convert division/day units (Villareal 1990) to per day units (our study).   We thus added in a sentence to the paper as follows:

 *"Notably, these low phytoplankton growth rates are in the same range as other diatoms with Richelia symbionts, namely 0.3 d$^{-1}$ for Rhizosolenia-Richelia cultured at a similar temperature (Villareal, 1990)."*

-Line 649 - This section (4.2) was a bit confusing, and I think the point could be made in much fewer words. Suggest revising for clarity and brevity.

We have shortened section 4.2 while still keeping the main point. We do think it is important to define compositional and aggregative variability since many readers may not be familiar with those terms so we left the beginning of the section unchanged. But near the end  before we went into more details about the differences among years, we have now omitted some sentences and kept discussion to a minimum.

Additional Figures  Below we show the figures we have added to the paper in case they are of interest to the reviewer.

[Figure]

Fig. 5. Box plots of data in the summer, mid-shelf region for a) chlorophyll associated with cells > 20 μm in units of mg m⁻³, b) net community production (NCP) and c) gross oxygen production (GOP) both in units of mmol $O_2$ m⁻² d⁻¹, d) NCP/GOP (unitless) which is a measure of export efficiency, e) Net Primary Production (NPP) in units of mg C m⁻² d⁻¹, f) silicate and g) phosphate, both in units of μmol L⁻¹, h) sea surface temperature in degrees Celsius and i) salinity in psu.. These plots show the differences in the plotted variables that occurred in August 2019 (orange box in each plot), a year when *Hemiaulus* carbon equaled 28.4 ug L⁻¹, compared to the data from the other summers, all of which had *Hemiaulus* carbon <0.02 μg L⁻¹.

[Figure]

Figure S1. Rates of net community production (NCP) and *Hemiaulus* carbon in the first half of the cruise (panels a and b) and the second half of the cruise (panels c and d). The mid-shelf region is circled for each panel and bathymetry contours are labeled in the first panel. Note that the later time period has smaller NCP and lower amounts of *Hemiaulus* carbon, and this, taken together with the near zero silicate values, suggests the bloom was likely in decline.

[Figure]

[Figure]

Figure S2. Figure 2. IFCB-based observations of carbon concentration associated with *Hemiaulus* collected across the NES broadscale cruise over the last decade emphasize the extreme nature of the high concentrations observed in 2019. (a) Map of automated IFCB measurement locations on 26 broadscale survey cruises conducted in the period 2013-2023 in partnership with NOAA's National Marine Fisheries Service, through their EcoMon program plus a couple of other ship-based observational programs (AMAPPS, HAB cyst surveys). IFCB sample locations (N = 19376) extend across the continental shelf from North Carolina to Maine, with the sampling distribution over the three 2019 cruises (N = 2253) highlighted by magenta coloring. (b) Normalized histograms of *Hemiaulus* carbon concentration in 2019 (magenta bars) and for all years except 2019 (blue bars). While most observations had undetectable *Hemiaulus* (0 mgC L$^{-1}$) in both periods (94.9% in 2019; 99.4% in all other years), the high concentration tail in 2019 is extraordinary compared to the rest of the decade.

---

## Author Comment (AC2)

Dear Editor,

We are thankful for the thoughtful reviews. Below we respond to Reviewer 2 in detail. Reviewer 2 suggested several new figures. We have included these figures at the end of this reply.

In the response below,

> Reviewer 2 text is in blue
> Our response is in black.
> *Direct quotes from the revised manuscript are in black italics and different font.*

Sincerely,
Rachel Stanley (on behalf of all the authors)

**Reviewer 2:** This study describes the novel observation of the diatom *Hemiaulus*, including biological fixing symbionts, blooming in the mid-shelf of coastal Northeastern United States during the summer of 2019. No information is provided about the biological nitrogen fixation activity of the diatom-diazotroph association (DDA) bloom. Instead, the authors focus on understanding: 1) the effect of the bloom in the production and transfer of phytoplankton organic matter, and 2) the environmental conditions responsible for the bloom formation. They use for that a broad compilation of physical, chemical and biological data obtained on multiple cruises within the framework of the NES-LTER program, together with information from remote sensing. The authors conclude that the DDA bloom led to increases in the production and export of phytoplankton organic matter and the transfer efficiency from phytoplankton to microzooplankton, whereas the environmental drivers responsible for the bloom formation remain unresolved.

The study addresses an interesting topic, as despite their ubiquitous nature and significance, DDA remain understudied and poorly understood. However, I have several main concerns about this study which prevent me for recommendation for publication, at least in the current form.

We appreciate the referee's thorough review of the manuscript, and have addressed each of their comments in detail below. We agree that the environmental drivers responsible for the observed *Hemiaulus* bloom remain unresolved– however we do not agree this should preclude publication. We humbly suggest that descriptive analysis of this phenomenon is scientifically valuable in and of itself, and sets the stage for more complete understanding in future studies, likely involving modeling of some sort.

**Specific comments**

1. **Data do not clearly support conclusions**. Based on the information collected in August 2019 the study shows how the DDA bloom was associated, in general, with higher total and large-sized fractionated chlorophyll, and increased net community production (Figure 1, 3, 4 and 8), compared to the inner and outer shelf. Then, the authors used climatological data (2018-2022) collected in the region (Table 2, Figures 5,6, 7 & 9) to conclude that the DDA bloom led to increases in net community production (NCP), gross oxygen production (GOP), net primary production (NPP), higher export efficiency, and higher trophic transfer efficiency from phytoplankton to microzooplankton.

However, this conclusion is based on the comparison of particular variables and individual years, and it is not evident when all the information is combined, as an important interannual (or derived from the comparison of different years) variability exists. Authors should consider using different statistical analysis and/or graphical visualization (as for example summer and winter box-plots for the inner, mid and outer shelf) in order to compare results from summer 2019 to the other years, and to convince the reader about the implications of the DDA bloom for the ecosystem functioning. The observation that other years (i.e 2018) also shows increases in Net Community Production despite different community composition and low chlorophyll concentration, questions the conclusion that the DDA bloom significantly impact productivity rates in the region.

We have made box plots as the reviewer suggested for NCP, GOP, NPP, export efficiency, for the mid-shelf region for all the summers from 2018 to 2022 (the figure is included at the end of this response and it is the new Fig. 5 in the manuscript). Additionally, we added boxplots for the surface Chl a, silicate, phosphate, temperature and salinity to further highlight the differences between the summers. Trophic transfer efficiency was already plotted as a bar plot in current Fig. 10 (old Fig. 9). There are only two grazing data points in the mid shelf region each summer so making a box plot of that data did not make sense. These plots show clearly that all the productivity metrics and Chl a associated with the large fraction are higher in summer 2019 than in any summer other than 2018, with the differences between 2019 and subsequent summers weakest for GOP but strong for the other summers  Thus, the summer of 2019 is different than the other summers. The plots also show that there is less of a difference for temperature and salinity between the summer of 2019 and the other summers. Since this study documents observations of a natural phenomenon rather than a controlled experiment,  there are multiple factors that are different between the summers. But taken together, the facts are: 2019 had much higher productivity metrics; 2019 was the only summer with observed *Hemiaulus* cells; there was remarkable coherence between NCP and the *Hemiaulus* abundance, as shown in Fig 9 (former fig 8); and there is logical coherence between having a DDA and increased production, strongly suggesting that the presence of *Hemiaulus* was directly responsible for the large NCP observed during the summer 2019 cruise . In particular, we think Fig. 9 (former Fig. 8) which uses data with resolution of several kilometers to show that at least 10 peaks in NCP are directly collocated with peaks in *Hemiaulus* carbon concentration is a "smoking gun" that shows at least for NCP, the presence of *Hemiaulus* is the reason why NCP is so large in 2019. None of the other productivity data are able to be measured on such high spatial resolution to show a direct correspondence.

It would have been fabulous if $N_2$ fixation rates had been measured on the cruise but they were not. We agree the story is complicated since 2018 also had high NCP and yet no *Hemiaulus* but the ocean is complex with multiple ways high productivity can be reached. The fact that NCP was also high in 2018 (when no *Hemiaulus* was observed and the Chl a associated with the size fraction was at typical summer levels) does not preclude *Hemiaulus* from being the cause of the high productivity in 2019. Additionally, the box plots clearly show how 2019 is anomalous in multiple productivity metrics. It would be fabulous if the time-series was longer and we had data prior to 2018 but we do not.

As well as including the box plots as the reviewer suggests, we changed the wording in multiple places throughout the paper to show that although the data suggests that *Hemiaulus* is causing

the high productivity, we cannot know for sure, especially for the metrics other than NCP (since NCP is the only data set at highest enough resolution to show such remarkable correspondence with *Hemiaulus* carbon concentration)

**Comparison of variables involving different time-space scales and limitations.** Several rates (NCP, GOP, NPP, growth rates) derived from different approaches are used to investigate the impact of the bloom in the production and transfer of organic matter. These approaches involve different temporal and spatial scales and methodological limitations. Therefore, the comparison of production rates derived from different approaches, specially when obtained in particular sampling dates (instead of the comparison of longer studies resolving for example seasonal scales) might be tricky. For example, is NPP derived from growth rates from dilution experiments comparable to NPP from 13C incorporation experiments? (Figure 7d). According to the methods section NCP rates computed in this study reflect the conditions over the past few days. Considering this relatively short period of time, could the NCP/GOP ratio be considered informative of the magnitude of carbon export efficiency? These facts could contribute to explain that the DDA bloom was characterized by high NCP but low phytoplankton growth rate, which despite the explanation included in the discussion section seems contradictory. These issues should be incorporated in the discussion.

We agree with the reviewer that the metrics of productivity reported here all have different time and space scales. However, we think this strengthens the conclusion that the summer of 2019 was fundamentally different from subsequent summers because of the presence of *Hemiaulus*. Since multiple metrics of productivity, which all have different assumptions and different temporal scales, point to the same conclusion that 2019 had anomalously large productivity, this is very likely a real change in the ecosystem rather than simply the consequence of a faulty assumption or a change on a fleeting time scale. In this paper, we are not directly comparing productivity rates to each other (comparing NPP rates to NCP for example) which could skew conclusions based on the different scales of the productivity methods. Rather we are comparing NPP from 2019 to NPP in 2020 – 2022, NCP in 2019 to NCP in 2018 – 2022, etc. We do use NCP and GOP together to calculate an export efficiency ratio as many papers have done previously since both NCP and GOP were derived from the same gas tracers and thus are directly comparable (see Juranek and Quay, 2012 for details). In related work, we have quantitatively compared the gas tracer and incubation-based metrics of productivity and found good correspondence but adding those comparisons to this paper will just lengthen and complicate the paper without improving it.

2.      **Unresolved formation mechanism:** Despite the discussion regarding the role of temperature and nutrients (based on phosphate and silicate concentrations, as nitrate was below the detection limit, which are not clearly different from the other years), the environmental conditions responsible for the bloom formation remain unresolved. The lack of information about nitrate concentrations and more important, nitrogen to phosphate ratios, clearly limits this task. The relevance of stratification is mentioned in the discussion, but no information is presented. The authors should consider including this variable in the analysis. They also mention that DDA bloom is associated with an ephemeral hydrographic feature which must require a different approach with higher temporal and spatial resolution. In this regard, animated versions of individual Chl a and SST images are NOT available at: http://science.whoi.edu/users/olga/outgoing/Aug_2019_chl/NEW_2019Hemiaulus/.

We agree with the reviewer that we have not been able to determine the reason why *Hemiaulus* bloomed in summer of 2019, but this diatom has not been observed in high numbers previously

on the Northeastern shelf (despite many regional cruises with IFCB measurements as part of NES-LTER and as part of the NOAA Ecomon program - see new figure S2, described in more detail below) or on any subsequent cruises. We documented in the manuscript what satellite data and reasonable oceanic explanations could provide and stated the unresolved questions. We think that the observations of the effects of the *Hemiaulus* bloom made in this cruise are still very exciting to the oceanographic community even though the mechanism for the origin of the bloom is still unclear. As the reviewer states, unfortunately the nitrate and ammonium data are below detection limits for all summers. This is not surprising – even with nitrogen fixation, all the fixed nitrogen produced is consumed in the upper layers and thus is not able to be detected with conventional autoanalyzer methods.

Stratification is only mentioned twice in the paper and is mentioned in general terms. Once, it is mentioned that, in this region, the summer is more stratified than the winter and several references are given to support that assertion (as well as it following from standard oceanographic knowledge). Secondly, in the conclusions, it is mentioned that stratification may increase with climate change. Stratification is not presented as an explanation of why *Hemiaulus* flourished in that particular summer and thus more detailed data on stratification would not add to the paper.

We thank the reviewer for alerting us that the link for the animated images no longer works. Rather than supplying another link that may also fail in the future, we have taken out reference to the animated images. The satellite data is still available at the site mentioned in the data sources at the end of the paper.

3.      **Messy structure and writing**: I find the structure and writing a bit messy, the arguments very speculative and hard to follow, as some references are missing, or necessary information is not included. For example, it is impossible to understand the conclusion that the bloom was decaying, as the methods section does not mention that the same geographical locations were occupied more than once. Structure and writing need revision.

Interestingly, Reviewer 1 found the writing very clear and the paper well-organized but we realize that everyone has different styles of reading and what works for one reader may  not work for another. Thus, we have tried to clarify the language in multiple places and also have added some details as requested. For example, in the methods section we now have added information about geographic locations being occupied more than once and thus the continuous data showing a progression in time (productivity incubations were only conducted once per station per cruise, even if a station was reoccupied):

"*The ship steamed both south and north along the longitude 70.88 W and thus over a  6 day cruise, the underway data sampled the same locations at multiple points in time. Stations were only occupied at one point in time per cruise. "*

We have also added a supplementary figure (new Fig. S1) that shows NCP and *Hemiaulus* abundance along the transect at the beginning and end of the cruise to support the conclusion that the bloom was decaying. (Note - we have attached Fig. S1 to the end of this response).

**Additional comments**

-I recommend starting the abstract by explaining the relevance of DDA, instead of general sentences about the relevance of productivity terms.

We have moved the first sentence of the abstract so the abstract does not start with the relevance of the productivity terms. With this edit, the statement about DDAs occurs earlier in the abstract.

-Introduction should include a description of the previous knowledge regarding seasonal variability (instead of the winter/summer comparison) of phytoplankton biomass activity and composition in the region.

Prior work (e.g. Marrec et al. 2021) has established that the contrasts between summer and winter with regards to environmental conditions, plankton community composition and biological activity are the most pronounced, so we chose to focus on this here. We prefer to leave the introduction as is, as we are not aiming for a description of the seasonality of the region but instead focus on the specific event of the *Hemiaulus* bloom.

-Lines 38-40 The Margalef´s mandala [*Margalef*, 1978] already mentioned the variety of life-strategies in diatoms.

We have added a reference to Margalef, 1978 .

-Incorporate the detailed description (time-space) of the cruises used in this study.

We have already given the time of each cruise in Table 1. The continuous data is collected throughout the entire time of the cruise. We have added to the text the exact latitudes which marked the beginning and end of the mid-shelf region (before it was described in terms of bathymetry and pointed out on a figure) and this change provides the location data for the mid-shelf underway data in general (the longitudes remain near constant as described in the paper). Providing every location for thousands of continuous data points would make the paper too long but the exact location of each data point is all in the published data which is available from the data links at the end of the paper. We have also added a supplementary table (Table S2) with the information on exact date and location of the mid-shelf stations on all the NES-LTER summer transect cruises. Additionally, we have marked the mid-shelf latitude on Fig. 1, 2, and 6-8.

-Modify the structure of the results section following: hydrography, nutrients, and finally biological parameters.

We appreciate this point and indeed we considered organizing it that way when we wrote it, but we want to mention the *Hemiaulus* distribution and Chl a results early in the paper since that explains why we focus on particular areas (mid-shelf) for the rest of the paper and also because the *Hemiaulus* abundances and associated productivity is the main point of the paper.

-Line 115 Cruise data are different from those in Table 1.

We have fixed the cruise date. The table and text now agree.

-Table 1 Include cruise track for all cruises, and available information for each one.
The table will be too long and complicated if one tries to show an entire cruise track on the table. Additionally, the cruise track is the same for all the NES-LTER cruises as we have clarified in the paper, so the information would be repetitive. The SPIROPA, OTZ and Ecomon cruises have different cruise tracks but the data used from SPIROPA and OTZ in this paper were in the exact same geographic location as NES-LTER – those cruises followed the same longitude/starting cruise track to allow collocation of sampling. The EcoMon cruise had a different cruise track and we have added in the geographical details for the EcoMon cruises as part of a new table S3 (which described temporal and geographical details for all Ecomon cruises since now multiple cruises are used in data for Fig. S2). Additionally, Ecomon data from August 2019 are shown on Fig. 11 (old Fig. 10) so the geographic location of that cruise is already in the paper. So instead of adding more detail to this table which would make it overly complicated, we clarified in the legend to the table and also in the paper that the data on Spiropa and OTZ is collocated with the NES-LTER data:

*"Cruise tracks for the NES-LTER cruises are given in Fig. 1. The SPIROPA and OTZ cruises followed the same longitude 70 53°W when in the mid-shelf region and thus data used from those cruises is collocated with the NES-LTER data."*

-Line 127 Need to specify how depths of the mixed layer, euphotic zone, and Chl-a maximum were calculated (better to explain it here instead in line 259-261).

We have moved the details on how the mixed layer was calculated. Note that the depth of the euphotic zone and the Chl max are not particularly relevant for this study since we are presenting all results in the mixed layer or integrated to the depth of the mixed layer. Nonetheless, we have added in details on how the euphotic zone and Chl max were determined.

*"Mixed layer depths were calculated from the temperature and salinity data from the CTD with the threshold method where the mixed layer was taken to be the depth where the density difference between the surface and bottom of the mixed layer was greater than $\Delta\sigma_\theta = 0.125$ kg m$^{-3}$ (de Boyer Montegut et al., 2004). Mixed layer depths were confirmed to be similar when a gradient criterion with a difference of 0.0125 kg m$^{-3}$ was used instead (Kara et al., 2000). Euphotic Zone was taken to be the depth at which light was 1% of the surface value. Chl a max was chosen based on the depth with maximum fluorescence observed in the CTD cast."*

-Detailed information about the variables used in this study is missing, for example sampling depths. Remove the description of those variable which are not used in this study (i.e ammonium).

Although we do not show data on ammonium and nitrate since values were below the detection limit, we still refer to those nutrients and the fact that they were below detection limits in the subsequent text. Thus we feel it makes sense to keep a short description of them in the methods section. The methods section describes all sampling and analysis methods for the other variables and gives references where even more detailed descriptions can be found.

-Lines 264-265. This sentence needs a reference to justify the used Chl-a to carbon ratio. Revise similar sentences through the manuscript where several references are missing.

We have added a reference and a more detailed explanation for the C:Chl ratio used. We have also added a few other references in the manuscript.

*"Primary production rates for 2018 were estimated from the growth/grazing rates. The surface values of phytoplankton growth rates were converted from Chl-a to carbon (mg C m$^{-3}$ d$^{-1}$) with a constant ratio of 50 which was then multiplied by the mixed layer depth to get values in mg C m$^{-2}$ d$^{-1}$. This calculated productivity from chlorophyll was then converted into NPP based on a linear relationship determined between Chl-calculated productivity and measured NPP during the summers when NPP rates were measured directly. While C:Chl ratios in coastal systems are highly variable seasonally (Jakobsen and Markager, 2016), we used the same C:Chl ratio when calculating the linear relationship and when scaling up NPP, thus the estimated NPP rates are insensitive to the choice of C:Chl ratio. C:Chl ratios were not used for NPP rate calculation for any other year."*

-Lines 363-364. Consider to incorporate *Hemiaulus* abundance data from previous years.

We already have averages of Hemiaulus carbon for all the summer NES-LTER transect cruises in Table 2. We considered also including a figure of Hemiaulus data from previous years in the new box and whisker plot but since the concentration in previous years is 100x smaller than the concentration is 2019, the plot wasn't very informative and we thought it better to include plots of other variables. Below is the plot we made but decided not to include. If the reviewer prefers, we could include it in the box plot (new Fig. 5, also attached at the end of this response) but then we would need to remove another panel, perhaps the panel on salinity.

[Figure]

We also have added a lot of data on *Hemiaulus* abundances in the region; we added *Hemiaulus* data from the EcoMon cruises from 2013-2023 in supplementary figure S2 in order to support the extraordinary nature of this bloom (figure is included at the end of this response). In the main text, we have added the following sentences near the beginning of section 3.1:

*"Hemiaulus carbon concentrations observed in other years on NES-LTER transect cruises never reached values above 0.30 µg L$^{-1}$, so approximately two orders of magnitude smaller than was observed on the 2019 cruise. Furthermore, IFCB-based observations made on a broader scale from the mid-Atlantic bight to the Gulf of Maine in the period from 2013 to 2023, show that only in August 2019 is Hemiaulus present in large quantities (Fig. S2), confirming the extraordinary nature of the 2019 bloom."*

-Lines 381-392. These sentences indicate that the same sampling transect was occupied more than once. No information is provided.

As described above, we have added information on the multiple occupations for the continuous data and also have included a figure in the supplementary material (new Fig. S1) that shows the data for the transect at the beginning and end of the cruise (figure is shown at the end of this response).

-Lines 394-406. I do not believe that the pattern observed in 3 out 5 years support this statement. In addition, Figure S2 shows important variability in temperature between July and August for the different years, which compromises the comparison made in Table 2 (comparison between years based on samplings carried out in different months).

The satellite data shows us that August does not in general have higher Chl a than July – in fact, it is often the reverse. Therefore, the higher Chl a observed in August associated with the *Hemiaulus* bloom is not solely due to cruise timing. Additionally, in order to strengthen the case that the *Hemiaulus* bloom was not typical for the area,  we have added in a supplementary figure (new Fig. S2 with cruise details in Table S3) of IFCB data from 26 Ecomon surveys throughout the NES region from 2013 to 2023. These data show that *Hemiaulus* was usually undetectable and never reached levels observed in August 2019 in any other year, even though numerous of the surveys from these other years were in August. We have added the figure itself to the supplementary material since the paper is already long. In the main manuscript, we have added the following text to replace some of the text that the reviewer did not think was supported:

*"In many of the summers (2018, 2021, and 2022), Chl-a in July was actually higher than in August, suggesting that the timing of the 2019 cruise (end of August instead of end of July) was not a factor in explaining the anomalously high productivity observed in August, 2019. If anything, the change in timing of the 2019 NES-LTER cruise would lead us to expect the Chl to be lower in August than in July and thus the high Chl a observed in August, 2019 is even more startling. Satellite data cannot be used to confirm the presence or absence of Hemiaulus in any of the other Augusts. However, IFCB data from NES broadscale NOAA EcoMon surveys from 2013 to 2023 many of which occurred in August, always show minimal presence of Hemiaulus suggesting the observed bloom in August 2019 was indeed extraordinary and not simply related to the timing of the 2019 LTER cruise (Fig. S2)."*

The fact that temperature is different in July and August is interesting but less important for explaining the *Hemiaulus* and the productivity rates in this study. The manuscript contains an entire section (Section 3.2) describing temperature changes and their associated variability and possible implications. Since the temperature we observed in August 2019 is similar to the temperature observed in the other cruises (even though they were earlier in the summer), it is

likely that temperature is not the cause of the higher productivity observed in August 2019. The new box plot we have added on temperature as part of Fig 5 shows that temperature in Aug 2019 is similar to other years. We have added some language going further into this.

-Lines 580-582. Speculation. Low values of phosphate, similar to August 2019, were sampled in July 2021.

That is true. This data set is very complex and sometimes there are similar conditions along one axis but not along others. For example, the difference in July 2021 as compared to Aug 2019 is that there were very low productivity rates (NCP, NPP, and GPP) when phosphate was low and also silicate was not drawn down. Thus July 2021 is a situation with low phosphate and low productivity whereas August 2019 is a system with low phosphate but high productivity. Our language in the paper reflects our admitted lack of certainty in the conclusions (the ocean is complicated) – it says "likely " and "suggests" rather than stating that any of it is absolutely true. Additionally, we have removed the word "only" from the paper and added some clarifications, such as:

*"While the summer of 2021 also had very low phosphate, summer 2021 was different in that it also had low productivity rates and more typical levels of silicate, suggesting the low phosphate occurred for fundamentally different reasons in 2019 and 2021."*

-Lines 586-587. No clear how the stage of the bloom was inferred from Figure 6.

Since silicate has been depleted in the surface waters and since *Hemiaulus* needs silicate, the bloom will have to be on the edge of declining or actively declining – there is no longer any silicate to maintain the bloom. We only have the nutrient data from one point in time. We have the continuous data at more times and now in the supplementary material, we show the continuous *Hemiaulus* and NCP data so readers can compare the beginning and end of the cruise (Fig. S1).

-Line 588. Recent studies suggest a unimodal relationship (instead inverse) between phytoplankton cell size and growth rate [*Marañón*, 2019].

We agree that there is a unimodal relationship between phytoplankton cell size and growth rates, as reported by Marañón (2015) (see figure from Maranon below). Indeed, maximum population growth rates of small picocyanobacteria (<1um) are expected to be in the same range (<0.4 d-1) as maximum growth rates of large phytoplankton (>20um) such as *Hemiaulus*, with population growth rates peaking for phytoplankton cells between 2um to 10um. Interestingly, when comparing division rates of *Synechococcus* and PicoEukaryote in the NES in summer (Fowler et al., 2020), division rates of PicoEukaryote were 2 to 3 times higher than those of smaller *Synechococcus*, which further support the expected unimodal relationship between phytoplankton cell size and growth rates (at the lower end of the size spectrum). It is worth noting that the phytoplankton community structure in summer in the NES is not dominated by picocyanobacteria (*Prochlorococcus* and *Synechococcus*), but rather by a combination of *Synechococcus*, picoeukaryote and nanoeukaryotes cells ( Fowler et al., 2020), which can be associated to the high growth rates observed the other summers. In addition, our low observed growth rates are similar to the rates "typical" for diatom-$N_2$ fixation associations obtained from

culture work of a different diatom with Rhizosolenia-Richelia symbionts (Villareal, 1990).

[Figure]
 (Figure from Maranon, 2015)

-Figures: Incorporate marks for differentiating inner, mid and outer shelf.

We have added a bar indicating the mid-shelf in each figure. We don't want to clutter the figures and since the inner shelf is always closer to shore and the outer shelf is further from shore (and the figures are made vs latitude so it is easy to tell what is closer and what is further from shore), we have chosen to only label the mid-shelf.

-Figure 3. Is the depth range 50-100 correct?

We removed mention of the depth range since it was confusing. The bloom was typically shallower than 50 m.

-Table 1. Incorporate information for variables available for each cruise,

The time for each cruise is given in the table already and the locations are the same for all the NES-LTER cruises and very similar for the others. Specific location data cannot easily fit into the table. So instead, we have clarified in the text that all the NES-LTER cruises have the same cruise track pictured and have explained in more detail the location of the non-NES-LTER cruises Additionally, we have added a supplementary table that has some details on exact times and locations for the NES-LTER transect cruises, which form most of the data presented in this paper.

-Table 2. Longitudinal range used for the mid-shelf should be specified, number of data (n) for averages should be indicated.

We have added latitudinal range to the caption. All data in the table was collected on the same longitude, as specified in the paper so a longitudinal range is not needed but we have added the longitude to the caption. We also have added the number of data used for the averages into the table. In order to make this fit (the table had many columns already), we have transposed the table.

Additional Figures  Below we show the figures we have added to the paper.

[Figure]

Fig. 5. Box plots of data in the summer, mid-shelf region fora) chlorophyll associated with cells > 20 μm in units of mg m$^{-3}$, b) net community production (NCP) and c) gross oxygen production (GOP) both in units of mmol O$_2$ m$^{-2}$ d$^{-1}$, d) NCP/GOP (unitless) which is a measure of export efficiency, e) Net Primary Production (NPP) in units of mg C m$^{-2}$ d$^{-1}$, f) silicate and g) phosphate, both in units of μmol L$^{-1}$, h) sea surface temperature in degrees Celsius and i) salinity in psu.. These plots show the differences in the plotted variables that occurred in August 2019 (orange box in each plot), a year when *Hemiaulus* carbon equaled 28.4 ug L$^{-1}$, compared to the data from the other summers, all of which had *Hemiaulus* carbon <0.02 μg L$^{-1}$.

[Figure]

Figure S1. Rates of net community production (NCP) and *Hemiaulus* carbon in the first half of the cruise (panels a and b) and the second half of the cruise (panels c and d). The mid-shelf region is circled for each panel and bathymetry contours are labeled in the first panel. Note that the later time period has smaller NCP and lower amounts of *Hemiaulus* carbon, and this, taken together with the near zero silicate values, suggests the bloom was likely in decline.

[Figure]

[Figure]

Figure S2. Figure 2. IFCB-based observations of carbon concentration associated with *Hemiaulus* collected across the NES broadscale cruise over the last decade emphasize the extreme nature of the high concentrations observed in 2019. (a) Map of automated IFCB measurement locations on 26 broadscale survey cruises conducted in the period 2013-2023 in partnership with NOAA's National Marine Fisheries Service, through their EcoMon program plus a couple of other ship-based observational programs (AMAPPS, HAB cyst surveys). IFCB sample locations (N = 19376) extend across the continental shelf from North Carolina to Maine, with the sampling distribution over the three 2019 cruises (N = 2253) highlighted by magenta coloring. (b) Normalized histograms of *Hemiaulus* carbon concentration in 2019 (magenta bars) and for all years except 2019 (blue bars). While most observations had undetectable *Hemiaulus* (0 mgC L$^{-1}$) in both periods (94.9% in 2019; 99.4% in all other years), the high concentration tail in 2019 is extraordinary compared to the rest of the decade.